# Plant photoreceptors and their signaling components compete for COP1 binding via VP peptide motifs

Kelvin Lau[1,†,‡] (ID), Roman Podolec[1,2,‡] (ID), Richard Chappuis[1], Roman Ulm[1,2,*] (ID) & Michael Hothorn[1,**] (ID)

## Abstract

Plants sense different parts of the sun's light spectrum using distinct photoreceptors, which signal through the E3 ubiquitin ligase COP1. Here, we analyze why many COP1-interacting transcription factors and photoreceptors harbor sequence-divergent Val-Pro (VP) motifs that bind COP1 with different binding affinities. Crystal structures of the VP motifs of the UV-B photoreceptor UVR8 and the transcription factor HY5 in complex with COP1, quantitative binding assays, and reverse genetic experiments together suggest that UVR8 and HY5 compete for COP1. Photoactivation of UVR8 leads to high-affinity cooperative binding of its VP motif and its photosensing core to COP1, preventing COP1 binding to its substrate HY5. UVR8–VP motif chimeras suggest that UV-B signaling specificity resides in the UVR8 photoreceptor core. Different COP1–VP peptide motif complexes highlight sequence fingerprints required for COP1 targeting. The blue-light photoreceptors CRY1 and CRY2 also compete with transcription factors for COP1 binding using similar VP motifs. Thus, our work reveals that different photoreceptors and their signaling components compete for COP1 via a conserved mechanism to control different light signaling cascades.

**Keywords** COP1; E3 ubiquitin ligase; light signaling; UV-B receptor; UVR8
**Subject Categories** Plant Biology
The EMBO Journal (2019) 38: e102140

See also: **Q Wang & C Lin** (September 2019)

## Introduction

Flowering plants etiolate in darkness, manifested by the rapid elongation of the embryonic stem, the hypocotyl, and closed and underdeveloped embryonic leaves, the cotyledons. Under light and upon photoreceptor activation, seedlings de-etiolate and display a photomorphogenic phenotype, characterized by a short hypocotyl and open green cotyledons, enabling a photosynthetic lifestyle (Gommers & Monte, 2018). The *constitutively photomorphogenic 1* (*cop1*) mutant displays a light-grown phenotype in the dark, including a short hypocotyl, and open and expanded cotyledons. *COP1* is thus a crucial repressor of photomorphogenesis (Deng *et al*, 1991). COP1 contains an N-terminal zinc finger, a central coiled-coil, and a C-terminal WD40 domain, which is essential for proper COP1 function (Deng *et al*, 1992; McNellis *et al*, 1994). Light-activated phytochrome, cryptochrome, and UVR8 photoreceptors inhibit COP1's activity (von Arnim & Deng, 1994; Hoecker, 2017; Podolec & Ulm, 2018). Although COP1 can act as a stand-alone E3 ubiquitin ligase *in vitro* (Saijo *et al*, 2003; Seo *et al*, 2003), it forms higher-order complexes *in vivo*, for example, with SUPPRESSOR OF PHYA-105 (SPA) proteins (Hoecker & Quail, 2001; Zhu *et al*, 2008; Ordoñez-Herrera *et al*, 2015). COP1 can also act as a substrate adaptor in CULLIN4–DAMAGED DNA BINDING PROTEIN 1 (CUL4-DDB1)-based heteromeric E3 ubiquitin ligase complexes (Chen *et al*, 2010). These different complexes may finetune COP1's activity toward different substrates (Ren *et al*, 2019). COP1 regulates gene expression and plays a central role as a repressor of photomorphogenesis by directly modulating the stability of transcription factors that control the expression of light-regulated genes (Lau & Deng, 2012; Podolec & Ulm, 2018). For example, the bZIP transcription factor ELONGATED HYPOCOTYL 5 (HY5) acts antagonistically with COP1 (Ang *et al*, 1998). COP1 binding to HY5 leads to its subsequent degradation via the 26S proteasome in darkness, a process that is inhibited by light (Osterlund *et al*, 2000).

In addition to HY5, other COP1 targets have been identified including transcriptional regulators, such as the HY5 homolog HYH (Holm *et al*, 2002), CONSTANS (CO) and other members of the BBX protein family (Jang *et al*, 2008; Liu *et al*, 2008; Khanna *et al*, 2009; Xu *et al*, 2016; Lin *et al*, 2018; Ordoñez-Herrera *et al*, 2018), and others such as LONG HYPOCOTYL IN FAR-RED 1 (HFR1) (Jang *et al*, 2005; Yang *et al*, 2005) and SHI-RELATED SEQUENCE 5 (SRS5) (Yuan *et al*, 2018). It has been suggested that specific Val-Pro (VP)-peptide motifs with a core sequence V-P-E/D-Φ-G, where Φ designated a hydrophobic residue, are able to bind the COP1

1 Department of Botany and Plant Biology, Section of Biology, Faculty of Sciences, University of Geneva, Geneva, Switzerland
2 Institute of Genetics and Genomics of Geneva (iGE3), University of Geneva, Geneva, Switzerland
*Corresponding author. Tel: +41-22-3793650; E-mail: roman.ulm@unige.ch
**Corresponding author. Tel: +41-22-3793013; E-mail: michael.hothorn@unige.ch
‡These authors contributed equally to this work
†Present address: Protein Production and Structure Core Facility, École Polytechnique Fédérale de Lausanne, Lausanne, Switzerland

WD40 domain (Holm *et al*, 2001; Uljon *et al*, 2016). Deletion of regions containing the VP peptide motifs results in the loss of interaction of COP1 substrates with the COP1 WD40 domain (Holm *et al*, 2001; Jang *et al*, 2005; Datta *et al*, 2006). Recently, it has been reported that the human COP1 WD40 domain directly binds VP motifs, such as the one from the TRIB1 pseudokinase that acts as a scaffold facilitating ubiquitination of human COP1 substrates (Uljon *et al*, 2016; Durzynska *et al*, 2017; Newton *et al*, 2018; Kung & Jura, 2019).

*Arabidopsis* photoreceptors for UV-B radiation (UV RESISTANCE LOCUS 8, UVR8), for blue light (cryptochrome 1 and 2, CRY1/CRY2) and for red/far-red light (phytochromes A–E), are known to repress COP1 activity in a light-dependent fashion (Yang *et al*, 2000, 2018a; Wang *et al*, 2001; Yu *et al*, 2007; Favory *et al*, 2009; Jang *et al*, 2010; Lian *et al*, 2011; Liu *et al*, 2011; Zuo *et al*, 2011; Viczián *et al*, 2012; Huang *et al*, 2013; Lu *et al*, 2015; Sheerin *et al*, 2015). UVR8 itself contains a conserved C-terminal VP peptide motif that is critical for UV-B signaling (Cloix *et al*, 2012; Yin *et al*, 2015). Moreover, overexpression of the UVR8 C-terminal 44 amino acids results in a *cop*-like phenotype (Yin *et al*, 2015). A similar phenotype has been observed when overexpressing the COP1-interacting CRY1 and CRY2 C-terminal domains (CCT; Yang *et al*, 2000, 2001). Indeed, CRY1 and CRY2 also contain potential VP peptide motifs within their CCT domains, but their function in blue-light signaling has not been established (Lin & Shalitin, 2003; Müller & Bouly, 2015). The presence of VP peptide motifs in different light signaling components suggests that COP1 may use a common targeting mechanism to interact with downstream transcription factors and upstream photoreceptors. Here, we present structural, quantitative biochemical, and genetic evidence for a VP peptide based competition mechanism, enabling COP1 to play a crucial role in different photoreceptor pathways in plants.

## Results

### The COP1 WD40 domain binds VP motifs from UVR8 and HY5

The WD40 domains of human and *Arabidopsis* COP1 directly interact with VP-containing peptides (Uljon *et al*, 2016). Such a VP peptide motif can be found in the UVR8 C-terminus that is not part of the UV-B-sensing β-propeller domain (Fig 1A; Kliebenstein *et al*, 2002; Rizzini *et al*, 2011; Christie *et al*, 2012; Wu *et al*, 2012), but is essential for UV-B signaling (Cloix *et al*, 2012; Yin *et al*, 2015). HY5 (Oyama *et al*, 1997), which is a COP1 target acting downstream of UVR8 in the UV-B signaling pathway (Ulm *et al*, 2004; Brown *et al*, 2005; Oravecz *et al*, 2006; Binkert *et al*, 2014), also contains a VP peptide motif (Fig 1A; Holm *et al*, 2001). UV-B absorption leads to UVR8 monomerization, COP1 binding, and subsequent stabilization of HY5 (Favory *et al*, 2009; Rizzini *et al*, 2011; Huang *et al*, 2013). Mutation of the HY5 VP pair to alanine (AA) stabilizes the HY5 protein (Holm *et al*, 2001).

In order to compare how the VP peptide motifs from different plant light signaling components bind COP1, we quantified the interaction of the UVR8 and HY5 VP peptides with the recombinant *Arabidopsis* COP1$^{349–675}$ WD40 domain (termed COP1 thereafter) using isothermal titration calorimetry (ITC). We find that both peptides bind COP1 with micromolar affinity and with HY5$^{39–48}$ binding ~ 8 times stronger than

UVR8$^{406–413}$ (Fig 1B). Next, we solved crystal structures of the COP1 WD40 domain–VP peptide complexes representing UVR8$^{406–413}$–COP1 and HY5$^{39–48}$–COP1 interactions, to 1.3 Å resolution (Fig 1C). Structural superposition of the two complexes (r.m.s.d. is ~ 0.2 Å comparing 149 corresponding $C_\alpha$ atoms) reveals an overall conserved mode of VP peptide binding (r.m.s.d is ~ 1.2 Å comparing six corresponding $C_\alpha$ atoms), with the central VP residues making hydrophobic interactions with COP1$^{Trp467}$ and COP1$^{Phe595}$ (buried surface area is ~ 500 Å$^2$ in COP1; Fig 1D and Appendix Fig S1). COP1$^{Lys422}$ and COP1$^{Tyr441}$ form hydrogen bonds and salt bridges with either UVR8$^{Tyr407}$ or HY5$^{Arg41}$, both being anchored to the COP1 WD40 core (Fig 1C and D, and Appendix Fig S1), as previously seen for the corresponding TRIB1 Gln356 residue in the COP1–TRIB1 peptide complex (Uljon *et al*, 2016). In our HY5$^{39–48}$–COP1 structure, an additional salt bridge is formed between HY5$^{Glu45}$ and COP1$^{His528}$ (Fig 1D). In the peptides, the residues surrounding the VP core adopt different conformations in UVR8 and HY5, which may explain their different binding affinities (Fig 1B and C). We tested this by mutating residues Lys422, Tyr441, and Trp467 in the VP peptide binding pocket of COP1. Mutation of COP1$^{Trp467}$ to alanine disrupts binding of COP1 to either UVR8- or HY5-derived peptides (Fig 1B and E). Mutation of COP1$^{Tyr441}$ to alanine abolishes binding of COP1 to the UVR8 peptide and greatly reduces binding to the HY5 peptide (Fig 1B and E), in good agreement with our structures (Fig 1D). The COP1$^{Lys422Ala}$ mutant binds HY5$^{39–48}$ as wild type, but increases the binding affinity of UVR8$^{406–413}$ ~ 10-fold (Fig 1B and E). Interestingly, COP1$^{Lys422Ala}$ interacts with full-length UVR8 also in the absence of UV-B in yeast two-hybrid assays, which is not detectable for wild type COP1 (Appendix Fig S2A; Rizzini *et al*, 2011). Moreover, COP1$^{Lys422Ala}$ also interacts more strongly with the constitutively interacting UVR8$^{C44}$ fragment (corresponding to the C-terminal UVR8 tail containing the VP motif) when compared to wild type COP1 in yeast two-hybrid assays (Appendix Fig S2B). In contrast, COP1$^{Tyr441Ala}$ and COP1$^{Trp467Ala}$ show reduced interaction to both UVR8 and HY5 (Appendix Fig S2). A UVR8$^{406–413}$–COP1$^{Lys422Ala}$ complex structure reveals the UVR8 VP peptide in a different conformation, with UVR8$^{Tyr407}$ binding at the surface of the VP-binding pocket (Appendix Fig S3A–E). In contrast, a structure of HY5$^{39–48}$–COP1$^{Lys422Ala}$ closely resembles the wild type complex (Appendix Fig S3F).

We next assessed the impact of COP1 VP peptide binding pocket mutants in UV-B signaling assays *in planta*. The seedling-lethal *cop1-5* null mutant can be complemented by expression of YFP-COP1 driven by the CaMV 35S promoter. We introduced COP1 mutations into this construct and isolated transgenic lines in the *cop1-5* background. All lines expressed the YFP-fusion protein and complemented the seedling lethality of *cop1-5* (Figs 1F and EV1A). We found that *cop1-5*/Pro$_{35S}$:YFP-COP1$^{Trp467Ala}$ and *cop1-5*/Pro$_{35S}$: YFP-COP1$^{Lys422Ala}$ transgenic lines have constitutively shorter hypocotyls when compared to wild type or *cop1-5*/Pro$_{35S}$:YFP-COP1 control plants (Fig 1F and G), in agreement with previous work (Holm *et al*, 2001), suggesting partially impaired COP1 activity. This is similar to the phenotype of *cop1-4* (Figs 1F and G, and EV1A–E), a weak *cop1* allele that is viable but fully impaired in UVR8-mediated UV-B signaling (McNellis *et al*, 1994; Oravecz *et al*, 2006; Favory *et al*, 2009). In contrast, *cop1-5*/Pro$_{35S}$:YFP-COP1$^{Tyr441Ala}$ showed an elongated hypocotyl phenotype when compared to wild type (Fig 1F and G), suggesting enhanced COP1 activity.

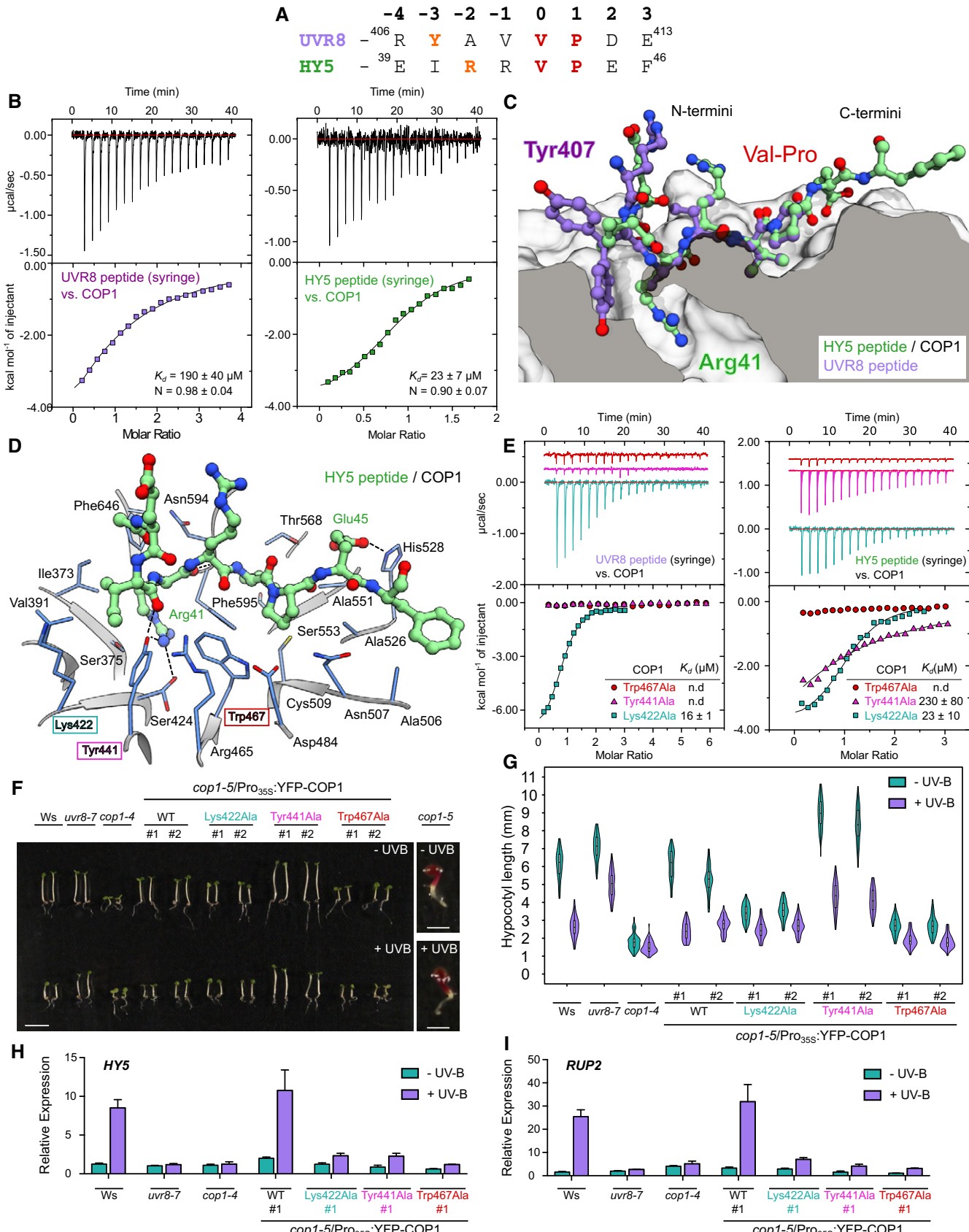

Figure 1.

◀

**Figure 1. The *Arabidopsis* COP1 WD40 domain binds peptides representing the core VP motifs of UVR8 and HY5 with different affinities.**

A   Alignment of the UVR8 and HY5 VP peptide motifs. The conserved VP pairs are highlighted in red and the anchor residues in orange.

B   ITC binding assays of the UVR8 and HY5 VP peptides versus the COP1 WD40 domain. The top panel represents the heats detected during each injection. The bottom panel represents the integrated heats of each injection, fitted to a one-site binding model (solid line). The following concentrations were typically used (titrant into cell): UVR8–COP1 (2,500 μM in 175 μM); HY5–COP1 (1,500 μM in 175 μM). The insets show the dissociation constants ($K_d$) and stoichiometries of binding (N) ($\pm$ standard deviation).

C   Superposition of the X-ray crystal structures of the HY5 and UVR8 peptides in the VP peptide binding site of the COP1 WD40 domain. COP1 is depicted in surface representation and belongs to the HY5–COP1 complex. The HY5 peptide is depicted in green in ball-and-stick representation (with Arg41 labeled). The UVR8 peptide from the UVR8–COP1 complex is superimposed on top in purple (with Tyr407 labeled), depicted in ball-and-stick representation. The surface of COP1 has been clipped to better visualize the anchor residue in the COP1 WD40 domain.

D   Ribbon diagram depicting the VP-binding site of COP1 (blue) bound to the HY5 peptide (green). Residues Lys422, Tyr441, and Trp467 are highlighted with a colored box in cyan, magenta, and red, respectively.

E   ITC assays of the HY5 and UVR8 VP peptides versus the different COP1 WD40 domain mutants (colors as in panel D). The following concentrations were typically used (titrant into cell): UVR8–COP1$^{Lys422Ala}$ (1,000 μM in 100 μM); UVR8–COP1$^{Tyr441Ala}$ (2,500 μM in 90 μM); UVR8–COP1$^{Trp467Ala}$ (2,500 μM in 90 μM); HY5–COP1$^{Lys422Ala}$ (1,500 μM in 175 μM); HY5–COP1$^{Tyr441Ala}$ (1,600 μM in 138 μM); and HY5–COP1$^{Trp467Ala}$ (1,600 μM in 112 μM). The insets show the dissociation constants ($K_d$). The stoichiometries of binding (N) for UVR8–COP1$^{Lys422Ala}$ = 0.87 $\pm$ 0.2; HY5–COP1$^{Lys422Ala}$ = 0.83 $\pm$ 0.2; and HY5–COP1$^{Tyr441Ala}$ = 0.92 $\pm$ 0.3 (all measurements $\pm$ standard deviation; n.d.: no detectable binding).

F, G   Images of representative individuals (F) and quantification of hypocotyl lengths (G) of 4-day-old seedlings grown with or without supplemental UV-B. The scale bars for all lines represent 5 mm except for *cop1-5* where the scale bars represent 1 mm. Violin and box plots are shown for $n > 60$ seedlings; upper and lower hinges correspond to the first and third quartiles; the horizontal line in the interior of the box indicates the median.

H, I   Quantitative real-time PCR analysis of (H) *HY5* and (I) *RUP2* expression. Four-day-old seedlings grown in white light were exposed to narrowband UV-B for 2 h (+ UV-B) or not (− UV-B). Error bars represent SEM of three biological replicates.

Data Information: In (F–I), lines used: wild type (Ws), *uvr8-7*, *cop1-4*, *cop1-5*/Pro$_{35S}$:*YFP-COP1* (WT), *cop1-5*/Pro$_{35S}$:*YFP-COP1*$^{Lys422Ala}$, *cop1-5*/Pro$_{35S}$:*YFP-COP1*$^{Tyr441Ala}$, *cop1-5*/Pro$_{35S}$:*YFP-COP1*$^{Trp467Ala}$, and *cop1-5*. #1 and #2: independent transgenic lines.

Importantly, in contrast to YFP-COP1, transcriptional responses for UV-B-induced marker genes like *HY5, RUP2, ELIP2,* and *CHS* are clearly abolished in YFP-COP1$^{Lys422Ala}$, YFP-COP1$^{Tyr441Ala}$, or YFP-COP1$^{Trp467Ala}$ after 2 h of UV-B treatment (Figs 1H and I, and EV1B and C). These responses represent an early read-out of UV-B signaling, which has been previously linked to UVR8-COP1-HY5 (Brown *et al*, 2005; Oravecz *et al*, 2006; Favory *et al*, 2009; Binkert *et al*, 2014). Surprisingly, however, the YFP-COP1$^{Lys422Ala}$ line showed strongly reduced UVR8 levels (Fig EV1A), despite showing normal *UVR8* transcript levels (Fig EV1F and G), precluding any conclusion of the mutation's effect on UV-B signaling *per se*. In contrast, YFP-COP1$^{Tyr441Ala}$ and YFP-COP1$^{Trp467Ala}$ were impaired in UV-B signaling, despite showing wild type UVR8 protein levels (Fig EV1A). This indicates strongly reduced UVR8 signaling, in agreement with the reduced affinity of the COP1 mutant proteins versus UVR8$^{406–413}$ *in vitro* (Fig 1E). Together, our crystallographic, quantitative biochemical, and functional assays suggest that UVR8 and HY5 can specifically interact with the COP1 WD40 domain using sequence-divergent VP motifs and that mutations in the COP1 VP-binding site can modulate these interactions and impair UVR8 signaling.

**High-affinity, cooperative binding of photoactivated UVR8**

HY5 levels are stabilized in a UVR8-dependent manner under UV-B light (Favory *et al*, 2009; Huang *et al*, 2013). We hypothesized that COP1 is inactivated under UV-B light, by activated UVR8 preventing HY5 from interacting with COP1. Our analysis of the isolated VP peptide motifs of UVR8 and HY5 suggests that UVR8 cannot efficiently compete with HY5 for COP1 binding. However, it has been previously found that the UVR8 β-propeller core can interact independently of its VP motif with the COP1 WD40 domain (Yin *et al*, 2015). We thus quantified the interaction of UV-B-activated full-length UVR8 with the COP1 WD40 domain. Recombinant UVR8 expressed in insect cells was purified to homogeneity, monomerized under UV-B, and analyzed in ITC and

grating-coupled interferometry (GCI) binding assays. We found that UV-B-activated full-length UVR8 binds COP1 with a dissociation constant ($K_d$) of ~ 150 nM in both quantitative assays (Fig 2A and B) and ~ 10 times stronger than non-photoactivated UVR8 (Appendix Fig S4A). This ~ 1,000-fold increase in binding affinity compared to the UVR8$^{406–413}$ peptide indicates cooperative binding of the UVR8 β-propeller core and the VP peptide motif. In line with this, UV-B-activated UVR8 monomers interact with the COP1 WD40 domain in analytical size-exclusion chromatography experiments, while the non-activated UVR8 dimer shows no interaction in this assay (Fig EV2A).

As the interaction of full-length UVR8 is markedly stronger than the isolated UVR8 VP peptide, we next dissected the contributions of the individual UVR8 domains to COP1 binding (Fig 2C). We find that the UV-B-activated UVR8 β-propeller core (UVR8$^{12–381}$) binds COP1 with a $K_d$ of ~ 0.5 μM and interacts with the COP1 WD40 domain in size-exclusion chromatography experiments (Fig EV2B and Appendix Fig S4B). The interaction is strengthened when the C-terminus is extended to include the VP peptide motif (UVR8$^{12–415}$; Appendix Fig S4B and C). Mutation of the UVR8 VP peptide motif to alanines results in ~ 20-fold reduced binding affinity when compared to the wild type protein (Fig 2D). However, the mutant photoreceptor is still able to form complexes with the COP1 WD40 domain in size-exclusion chromatography assays (Fig EV2C). We could not detect sufficient binding enthalpies to monitor the binding of UVR8$^{ValPro/AlaAla}$ to COP1 in ITC assays nor detectable signals in GCI experiments in the absence of UV-B (Appendix Fig S5). The COP1$^{Lys422Ala}$ mutant binds UV-B-activated full-length UVR8 with wild type affinity, while COP1$^{Trp467Ala}$ binds ~ 5 times more weakly (Appendix Fig S6A and B). Mutations targeting both COP1 and the UVR8 C-terminal VP peptide motif decrease their binding affinity even further (Appendix Fig S6C). Thus, full-length UVR8 uses both its β-propeller photoreceptor core and its C-terminal VP peptide to cooperatively bind the COP1 WD40 domain when activated by UV-B light.

We next asked whether UV-B-activated full-length UVR8 could compete with HY5 for binding to COP1. We produced the full-length HY5 protein in insect cells and found that it binds the COP1 WD40 domain with a $K_d$ of ~ 1 μM in GCI assays (Fig 2E). For comparison, the isolated HY5 VP peptide binds COP1 with a $K_d$ of ~ 20 μM (Fig 1B). This would indicate that only the UV-B-activated UVR8

and not ground-state UVR8 ($K_d$ ~ 150 nM versus ~ 1 μM, see above) can efficiently compete with HY5 for COP1 binding. We tested this hypothesis in yeast 3-hybrid experiments. We confirmed that HY5 interacts with COP1 in the absence of UVR8 and that this interaction is specifically abolished in the presence of UVR8 and UV-B light (Fig 2F). We conclude that UV-B-activated UVR8

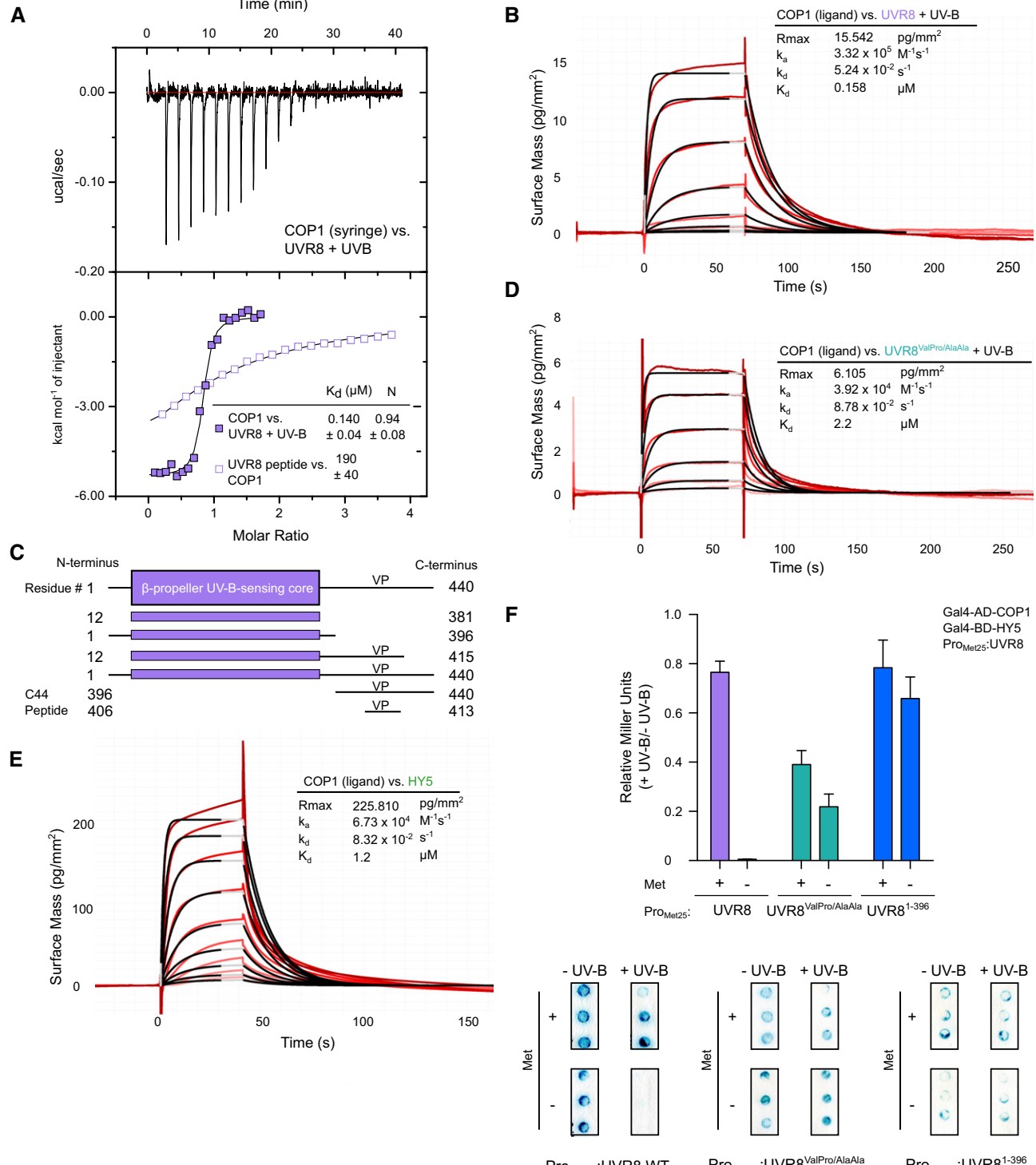

**Figure 2.**

efficiently competes with HY5 for COP1 binding in yeast cells, thereby impairing the COP1–HY5 interaction under UV-B. The UVR8[ValPro/AlaAla] and UVR8[1–396] mutants cannot interfere with the COP1–HY5 interaction in yeast cells (Fig 2F), suggesting that a functional UVR8 VP peptide motif is required to compete off HY5 from COP1, in agreement with our biochemical assays.

### UVR8–VP peptide chimeras trigger UV-B signaling *in planta*

Our findings suggest that UVR8 requires both its UV-B-sensing core and its VP peptide motif for high-affinity COP1 binding and that the UVR8 VP peptide can inhibit the interaction of HY5 with COP1 (Figs 1 and 2; Yin *et al*, 2015). This led us to speculate that any VP peptide with sufficient binding affinity for COP1 could functionally replace the endogenous VP motif in the UVR8 C-terminus *in vivo*. We generated chimeric proteins in which the UVR8 core domain is fused to VP-containing sequences from plant and human COP1 substrates, namely HY5 and TRIB1 (Fig 3A). *Arabidopsis uvr8-7* null mutants expressing these chimeric proteins show complementation of the hypocotyl and anthocyanin phenotypes under UV-B, suggesting that all tested UVR8 chimeras are functional (Figs 3B–D, and EV3). Early UV-B marker genes are also up-regulated in the lines after UV-B exposure, demonstrating that these UVR8 chimeras are functional photoreceptors, although to different levels (Fig 3E). In line with this, the UVR8[HY5C44] chimera can displace HY5 from COP1 in yeast 3-hybrid assays (Fig 3F). All the UVR8 chimeras can bind COP1 with affinities comparable to wild type (Figs 3G and EV3) and are dimers *in vitro* that monomerize under UV-B light (Fig 3H and Appendix Fig S7D). Together, these experiments reinforce the notion that divergent VP peptide motifs compete with each other for binding to the COP1 WD40 domain.

### Sequence-divergent VP peptide motifs are recognized by the COP1 WD40 domain

Our protein engineering experiments prompted us to map core VP peptide motifs in other plant light signaling components, including the COP1-interacting blue-light photoreceptors CRY1 and CRY2 (Yang *et al*, 2000, 2018a; Wang *et al*, 2001; Yu *et al*, 2007) and the transcription factors HYH, CO/BBX1, COL3/BBX4, SALT TOLERANCE (STO/BBX24) (Holm *et al*, 2002; Datta *et al*, 2006; Jang *et al*, 2008; Liu *et al*, 2008; Yan *et al*, 2011), and HFR1 (Duek *et al*, 2004; Jang *et al*, 2005; Yang *et al*, 2005). We mapped putative VP motifs in all these proteins and assessed their binding affinities to the COP1 WD40 domain (Fig 4A and B). We could detect binding for most of the peptide motifs in ITC assays, with dissociation constants in the mid-micromolar range (Figs 4A and EV4). Next, we obtained crystal structures for the different peptides bound to COP1 (1.3–2.0 Å resolution, see Tables EV1 and EV2, Figs 4E and F, and EV4) to compare their peptide binding modes (Fig 4C). We found that all peptides bind in a similar configuration with the VP forming the center of the binding site (r.m.s.d. between the different peptides range from ~ 0.3 to 1.5 Å, comparing 5 or 6 corresponding $C_\alpha$ atoms). Chemically diverse amino acids (Tyr/Arg/Gln) map to the −3 and −2 position and often deeply insert into the COP1 binding cleft, acting as anchor residues (Fig 4C). This suggests that the COP1 WD40 domain has high structural plasticity, being able to accommodate sequence-divergent VP-containing peptides.

To investigate this property of COP1, we quantified the interaction of different VP peptides with our COP1[Lys422Ala] mutant protein. As for UVR8 (Fig 1B and E), COP1[Lys422Ala] showed increased binding affinity for some peptides such as those representing the COL3[287–294] and CO[366–373] VP motifs, while it reduced binding to others, such as to CRY1[544–552] and CRY2[527–535] (Fig 4A and D). These observations may be rationalized by an enlarged VP-binding pocket in the COP1[Lys422Ala] mutant, increasing accessibility for the COL3[Phe288] anchor residue, and potentially abolishing interactions with CRY1[Asp545] (Fig 4E and F). In yeast 3-hybrid assays, we find that, similar to HY5 (Fig 2F), UV-B-activated UVR8 can efficiently compete with HYH and an N-terminal fragment of HFR1 for binding to COP1 (Fig EV5). Taken together, VP peptide motifs of cryptochrome photoreceptors and diverse COP1 transcription factor targets all bind to the COP1 WD40 domain and UVR8 is able to compete with COP1 partners for binding.

◀ **Figure 2. High-affinity cooperative binding of activated full-length UVR8 to the COP1 WD40 domain is mediated by its UV-B-activated β-propeller core and its C-terminal VP peptide motif.**

A  ITC assay between the COP1 WD40 domain and full-length UVR8 pre-monomerized by UV-B light. Integrated heats are shown in solid, purple squares. For comparison, an ITC experiment between UVR8 VP peptide and the COP1 WD40 domain (from Fig 1B) is also shown in open, purple squares. The following concentrations were typically used (titrant into cell): COP1–UVR8 + UV-B (130 μM in 20 μM). The inset shows the dissociation constant ($K_d$), stoichiometry of binding (N) (± standard deviation).

B  Binding kinetics of UVR8 pre-monomerized by UV-B versus the COP1 WD40 domain obtained by grating-coupled interferometry (GCI). Sensorgrams of UVR8 injected are shown in red, with their respective 1:1 binding model fits in black. The following amounts were typically used: ligand—COP1 (2,000 pg/mm$^2$); analyte—UVR8 + UV-B (highest concentration 2 μM). $k_a$ = association rate constant, $k_d$ = dissociation rate constant, $K_d$ = dissociation constant.

C  The domain organization of *Arabidopsis* UVR8. It consists of a UV-B-sensing β-propeller core (residues 12–381) and a long C-terminus containing the VP motif. Constructs and peptides used and their residue endings are indicated.

D  Binding kinetics of UVR8[ValPro/AlaAla] pre-monomerized by UV-B versus the COP1 WD40 domain obtained by GCI experiments. Sensorgrams of UVR8 injected are shown in red, with their respective 1:1 binding model fits in black. The following amounts were typically used: ligand—COP1 (2,000 pg/mm$^2$); analyte—UVR8[ValPro/AlaAla] +UV-B (highest concentration 2 μM). $k_a$ = association rate constant, $k_d$ = disassociation rate constant, $K_d$ = dissociation constant.

E  Binding kinetics of HY5 versus the COP1 WD40 domain obtained by GCI experiments. Sensorgrams of HY5 injected are shown in red, with their respective 1:1 binding model fits in black. The following amounts were typically used: ligand—COP1 (2,000 pg/mm$^2$); analyte—HY5 (highest concentration 2 μM). $k_a$ = association rate constant, $k_d$ = dissociation rate constant, $K_d$ = dissociation constant.

F  Yeast 3-hybrid analysis of the COP1–HY5 interaction in the presence of UVR8. (Top) Normalized Miller Units were calculated as a ratio of β-galactosidase activity in yeast grown under UV-B (+ UV-B) versus yeast grown without UV-B (− UV-B). Additionally, normalized Miller Units are reported separately here for yeast grown on media without or with 1 mM methionine, corresponding to induction (− Met) or repression (+ Met) of *Met25* promoter-driven UVR8 expression, respectively. Means and SEM for three biological repetitions are shown. (Bottom) representative filter lift assays. AD, activation domain; BD, DNA binding domain; Met, methionine.

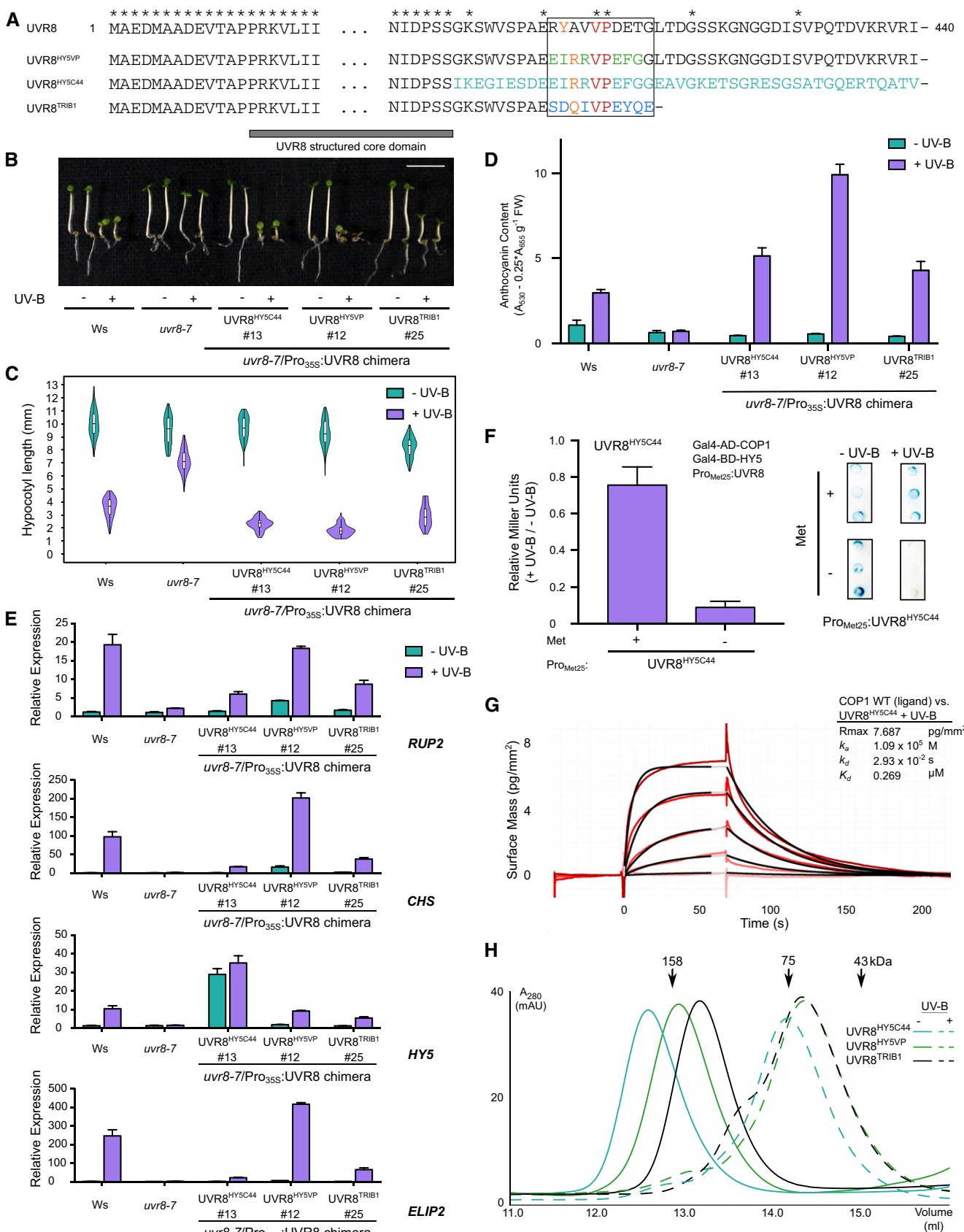

**Figure 3.**

◀

**Figure 3.  Chimeras between the UV-B-sensing UVR8 core and various VP motifs are functional in the UV-B signaling pathway.**

A   Sequence alignment of the N- and C- termini of UVR8, chimera UVR8^HY5VP (replacing the UVR8 VP motif with the corresponding sequence from HY5), chimera UVR8^HY5C44 (replacing the C44 domain of UVR8 with the corresponding sequence from HY5), and chimera UVR8^TRIB1VP (replacing the UVR8 VP motif with the TRIB1 VP motif and a truncation of the rest of the UVR8 C-terminus). The black box indicates the core VP motif of UVR8. The VP is colored in red, the anchor residues in orange, the divergent residues of the HY5 VP core sequence in green, the HY5 sequence replacing the UVR8 C-terminal 44 amino acids (C44) in cyan, and the divergent residues of the TRIB1 VP core sequence in blue. Asterisks represent amino acids identical in all constructs, and amino acids 21–390 of UVR8 are not shown. The previously crystallized UVR8 core domain (PDB: 4D9S) is highlighted with a gray bar.

B   Representative image showing the phenotype of wild type (Ws), *uur8-7* and *uur8-7/*Pro$_{35S}$:*UVR8*^HY5C44, *uur8-7/*Pro$_{35S}$:*UVR8*^HY5VP, and *uur8-7/*Pro$_{35S}$:*UVR8*^TRIB1 seedlings grown for 4 days under white light (− UV-B) or white light supplemented with UV-B (+ UV-B). The scale bar represents 5 mm.

C   Quantification of hypocotyl length data shown in (B). Violin and box plots are shown for $n > 60$ seedlings; upper and lower hinges correspond to the first and third quartiles; the horizontal line in the interior of the box indicates the median.

D   Anthocyanin accumulation in seedlings shown in B. Average and SEM are shown ($n = 3$).

E   Quantitative real-time PCR analysis of *RUP2*, *CHS*, *HY5*, and *ELIP2* expression in wild type (Ws), *uur8-7* and *uur8-7/*Pro$_{35S}$:*UVR8*^HY5C44, *uur8-7/*Pro$_{35S}$:*UVR8*^HY5VP, and *uur8-7/*Pro$_{35S}$:*UVR8*^TRIB1 4-day-old seedlings grown under white light in response to 2 h of UV-B (+ UV-B versus − UV-B). Error bars represent SEM of three biological replicates. Note: The primers used to detect the *HY5* transcript abundance also bind to an identical region present in UVR8^HY5C44 chimera.

F   Yeast 3-hybrid analysis of the COP1-HY5 interaction in the presence of the UVR8^HY5C44 chimera. (Left panel) Normalized Miller Units were calculated as a ratio of β-galactosidase activity in yeast grown under UV-B (+ UV-B) versus yeast grown without UV-B (− UV-B). Additionally, normalized Miller Units are reported separately here for yeast grown on media without or with 1 mM methionine, corresponding to induction (− Met) or repression (+ Met) of *Met25* promoter-driven UVR8 expression, respectively. Means and SEM for three biological repetitions are shown. (Right panel) Representative filter lift assays of the yeast analyzed in left panel. AD, activation domain; BD, DNA binding domain; Met, methionine.

G   Binding kinetics of the UVR8^HY5C44 chimera pre-monomerized by UV-B versus the COP1 WD40 domain obtained by GCI experiments. Sensorgrams of UVR8^HY5C44 injected are shown in red, with their respective 1:1 binding model fits in black. The following amounts were typically used: ligand—COP1 (2,000 pg/mm$^2$); analyte—UVR8^HY5C44 +UV-B (highest concentration 2 μM). $k_a$ = association rate constant, $k_d$ = disassociation rate constant, $K_d$ = dissociation constant.

H   Size-exclusion chromatography assay of purified, recombinant chimeric proteins expressed in Sf9 insect cells in the presence and absence of UV-B.

## CRY2 and CONSTANS compete for COP1 binding

The structural plasticity of the COP1 WD40 domain is illustrated by the variable modes of binding for sequence-divergent VP motifs found in different plant light signaling components. The COP1^Lys422Ala mutation can modulate the interaction with different VP peptide (Fig 4A). We noted that the *cop1-5/*Pro$_{35S}$:*YFP-COP1*^Lys422Ala but not other COP1 mutants show delayed flowering when grown in long days (Fig 5A–D). This phenotype has been previously associated with mutant plants that lack the COP1 substrate CO (Fig 5B–D; Putterill *et al*, 1995; Jang *et al*, 2008; Liu *et al*, 2008).

We thus hypothesized that in COP1^Lys422Ala plants, binding and subsequent degradation of CO may be altered under long-day conditions. *In vitro*, we found that the CO VP peptide binds COP1^Lys422Ala ∼ 4 times stronger than wild type COP1 (Fig 5F). The same mutation in COP1 strongly reduces (∼ 30 times) binding of the CRY2 VP peptide *in vitro* (Fig 5F). It is of note that, in contrast to UVR8 (Fig 1F), CRY2 levels are not altered in the COP1^Lys422Ala background (Fig 5E). Thus, the late-flowering phenotype of the COP1^Lys422Ala mutant suggests that CRY2 and CO compete for COP1 binding and that this competition is altered in the COP1^Lys422Ala mutant background: reduced affinity to CRY2, enhanced binding to CO—both consistent with the late-flowering phenotype. In line with this, we find that only recombinant light-activated full-length CRY2 binds wild type COP1 with nanomolar affinity in quantitative GCI experiments (Fig 5G and Appendix Fig S8). This ∼ 200-fold increase in binding affinity over the isolated CRY2 VP peptide strongly suggests that UVR8 and CRY2 both use a cooperative binding mechanism to target COP1. As a control, we tested a fragment of the CRY2 C-terminus containing the VP motif, the NC80 domain (CRY2^486–565; Yu *et al*, 2007). We found that NC80 binds COP1 with an affinity comparable to the isolated CRY2^527–535 VP peptide assayed by ITC (Fig 5F and H). Together, the COP1^Lys422Ala phenotypes and our

biochemical assays suggest that different plant photoreceptors may use a light-induced cooperative binding mechanism, preventing COP1 from targeting downstream light signaling partners for degradation.

## Discussion

The COP1 E3 ubiquitin ligase is a central signaling hub that integrates inputs from plant light-sensing photoreceptors. There is strong evidence that the UV-B-sensing photoreceptor UVR8, the blue-light receptors CRY1 and CRY2, and the red/far-red discriminating phytochromes all regulate COP1 activity (Hoecker, 2017; Podolec & Ulm, 2018). The regulation of COP1 by photoreceptors enables a broad range of photomorphogenic responses, including de-etiolation, cotyledon expansion, and transition to flowering, as well as UV-B light acclimation (Lau & Deng, 2012; Jenkins, 2017; Yin & Ulm, 2017; Gommers & Monte, 2018). Here, we have dissected at the structural, biochemical, and genetic levels how the activated UVR8 and cryptochrome photoreceptors impinge on COP1 activity, by interacting with its central WD40 domain, resulting in the stabilization of COP1 substrate transcription factors. For both types of photoreceptors, interaction through a linear VP peptide motif and a folded, light-regulated interaction domain leads to cooperative, high-affinity binding of the activated photoreceptor to COP1. We propose that in response to UV-B light, UVR8 dimers monomerize, exposing a new interaction surface that binds to the COP1 WD40 domain and releases the UVR8 C-terminal VP motif from structural restraints that prevent its interaction with COP1 in the absence of UV-B (Yin *et al*, 2015; Heilmann *et al*, 2016; Camacho *et al*, 2019; Wu *et al*, 2019). Similarly, the VP motif in the CCT domain of cryptochromes may become exposed and available for interaction upon blue-light activation of the photoreceptor (Müller & Bouly, 2015; Wang *et al*, 2018). Because UVR8 and CRY2 are very

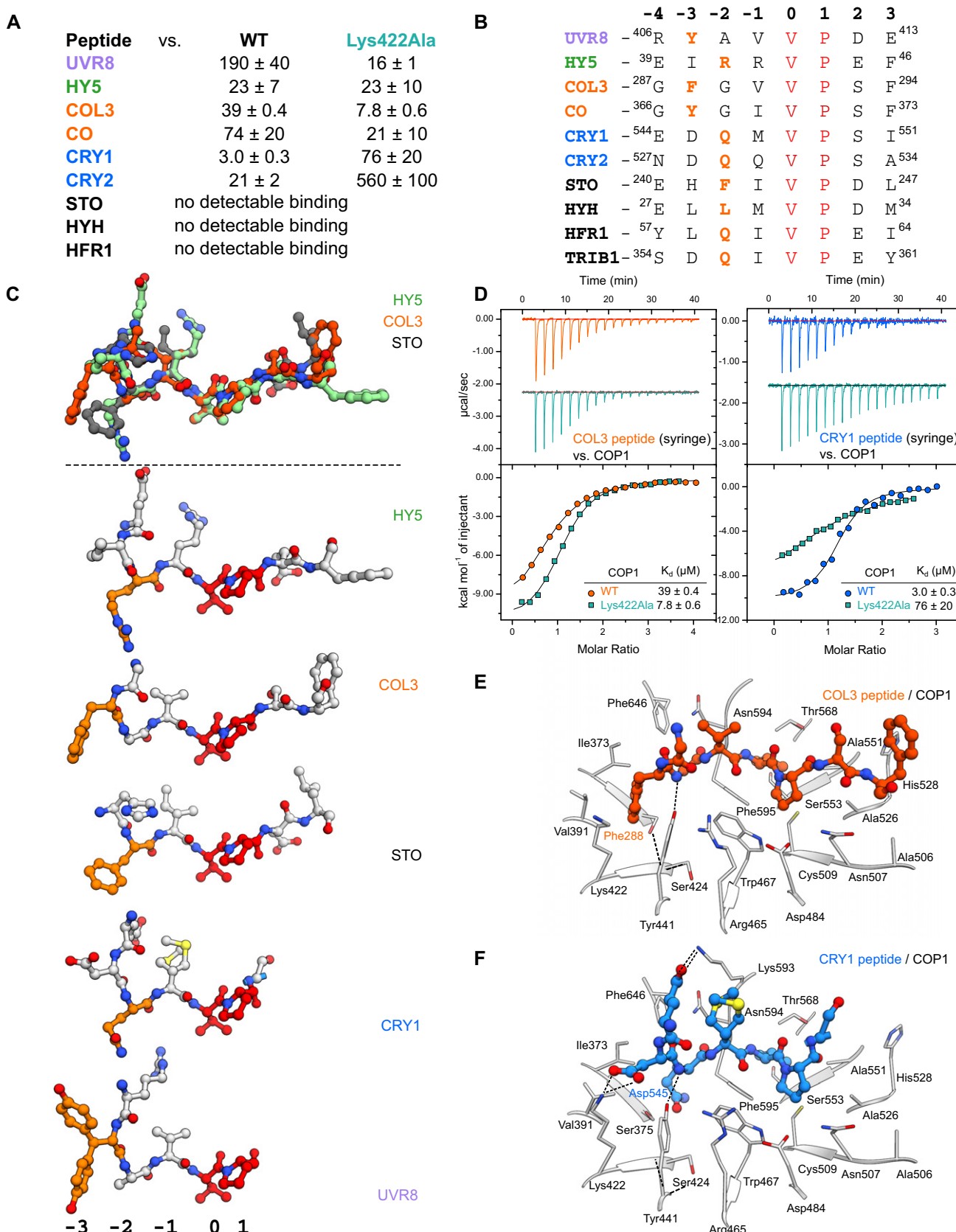

**A**

| Peptide | vs. | WT | Lys422Ala |
|---|---|---|---|
| UVR8 | | 190 ± 40 | 16 ± 1 |
| HY5 | | 23 ± 7 | 23 ± 10 |
| COL3 | | 39 ± 0.4 | 7.8 ± 0.6 |
| CO | | 74 ± 20 | 21 ± 10 |
| CRY1 | | 3.0 ± 0.3 | 76 ± 20 |
| CRY2 | | 21 ± 2 | 560 ± 100 |
| STO | | no detectable binding | |
| HYH | | no detectable binding | |
| HFR1 | | no detectable binding | |

**B**

| | | –4 | –3 | –2 | –1 | 0 | 1 | 2 | 3 | |
|---|---|---|---|---|---|---|---|---|---|---|
| UVR8 | – 406R | Y | A | V | V | P | D | E | 413 |
| HY5 | – 39E | I | R | R | V | P | E | F | 46 |
| COL3 | – 287G | F | G | V | V | P | S | F | 294 |
| CO | – 366G | Y | G | I | V | P | S | F | 373 |
| CRY1 | – 544E | D | Q | M | V | P | S | I | 551 |
| CRY2 | – 527N | D | Q | Q | V | P | S | A | 534 |
| STO | – 240E | H | F | I | V | P | D | L | 247 |
| HYH | – 27E | L | L | M | V | P | D | M | 34 |
| HFR1 | – 57Y | L | Q | I | V | P | E | I | 64 |
| TRIB1 | – 354S | D | Q | I | V | P | E | Y | 361 |

**C** HY5 / COL3 / STO / HY5 / COL3 / STO / CRY1 / UVR8

**D** COL3 peptide (syringe) vs. COP1

| COP1 | $K_d$ (µM) |
|---|---|
| WT | 39 ± 0.4 |
| Lys422Ala | 7.8 ± 0.6 |

CRY1 peptide (syringe) vs. COP1

| COP1 | $K_d$ (µM) |
|---|---|
| WT | 3.0 ± 0.3 |
| Lys422Ala | 76 ± 20 |

**E** COL3 peptide / COP1

**F** CRY1 peptide / COP1

**Figure 4.**

**Figure 4.  Many COP1 substrates and interacting photoreceptors contain VP peptides that can bind the COP1 WD40 domain.**

A   Table summarizing affinities ($K_d$, dissociation constant) of VP-containing peptides versus the COP1 WD40 domain (WT) and COP1$^{Lys422Ala}$ as determined by ITC. All values are µM ($\pm$ standard deviation).

B   A sequence comparison of VP peptide motifs that interact with the COP1 WD40 domain. The VP core is colored in red and the anchor residues in orange. The peptide sequences are numbered on a register where the V in the VP is 0.

C   (Top panel) Superposition of the VP peptide binding modes of the HY5 (green), COL3 (orange), and STO (gray) peptides depicted in ball-and-stick representation when bound to the COP1 WD40 domain as observed in their respective X-ray crystal structures. (Bottom panel) A comparison of the HY5, COL3, STO, CRY1, and UVR8 peptides highlights the chemically diverse anchor residues (orange) and their variant positions.

D   ITC assays between the COL3 VP peptide (left) and CRY1 VP peptide (right) versus the COP1 WD40 and COP1$^{Lys422Ala}$, with a table summarizing their corresponding affinities. The following concentrations were typically used (titrant into cell): COL3–COP1 (2,240 µM in 195 µM); COL3–COP1$^{Lys422Ala}$ (1,500 µM in 175 µM); CRY1–COP1 (750 µM in 75 µM); and CRY1–COP1$^{Lys422Ala}$ (1,500 µM in 175 µM). The inset shows the dissociation constant ($K_d$). The stoichiometries of binding (N) for COL3–COP1 = 0.75 $\pm$ 0.2; COL3–COP1$^{Lys422Ala}$ = 0.90 $\pm$ 0.2; CRY1–COP1 = 0.99 $\pm$ 0.3; CRY1–COP1$^{Lys422Ala}$ = 0.98 $\pm$ 0.1 (all measurements $\pm$ standard deviation).

E, F   The crystal structure of the (E) COL3 peptide and (F) CRY1 peptide bound to the COP1 WD40 domain. The peptides are depicted in ball-and-stick representation. Selected residues from COP1 are depicted in gray in stick representation.

different in structure and domain composition, they likely use distinct interaction surfaces to target the COP1 WD40 domain, in addition to the VP peptide motifs. The cooperative, high-affinity mode of binding enables UVR8 and cryptochromes to efficiently displace downstream signaling components such as HY5, HYH, HFR1, and CO in a light-dependent manner. Structure-guided mutations in the COP1 WD40 binding cleft resulted in the identification of the COP1$^{Lys422Ala}$ mutant, which displays flowering phenotypes, and COP1$^{Tyr441Ala}$ and COP1$^{Trp467Ala}$, which shows UV-B signaling phenotypes, that are all consistent with our competition model. Similar mutations have previously been shown to affect hypocotyl elongation in white light (Holm *et al*, 2001). It is interesting to note that the COP1$^{Tyr441Ala}$ mutant still shows UV-B-dependent hypocotyl growth inhibition even though the COP1$^{Tyr441Ala}$ protein is impaired in its interaction with UVR8. This UV-B response is possibly due to a recently described COP1-independent UVR8 activity involving direct interaction and inhibition of transcription factors that promote hypocotyl elongation (Liang *et al*, 2018; Yang *et al*, 2018b). Moreover, we cannot exclude that the UVR8 interaction with COP1 that is independent of the UVR8 C-terminal VP domain (Fig 2D, Appendix Fig S4B and Fig EV2B; see also Yin *et al*, 2015) can affect COP1$^{Tyr441Ala}$ activity sufficiently to see a long-term phenotype such as hypocotyl growth inhibition. Likewise, partial complementation of COP1$^{Trp467Ala}$ reveals that it may not be impaired in interacting with all COP1 substrates and that it may still target other

photomorphogenesis-promoting factors for degradation or retains an activity that is independent of the VP peptide binding pocket. Unexpectedly, COP1$^{Lys422Ala}$ rendered the UVR8 protein unstable, preventing conclusive analysis of the effect of this COP1 mutant on UV-B signaling *in vivo*. Moreover, the mechanism behind UVR8 protein instability remains to be determined. Independent of this, it is interesting to note that the *hy4-9* mutant, which replaces the proline in the CRY1 VP peptide motif with leucine, does not show inhibition of hypocotyl elongation under blue light (Ahmad *et al*, 1995). Similarly, mutations of the UVR8 VP peptide motif or C-terminal truncations (including the *uvr8-2* allele, which has a premature stop codon at Trp400) all strongly impair UV-B signaling (Brown *et al*, 2005; Cloix *et al*, 2012; Yin *et al*, 2015). We now report quantitative biochemical and crystallographic analyses that reveal that UVR8 and cryptochrome photoreceptors and their downstream transcription factors all make use of VP-containing peptide motifs to target a central binding cleft in the COP1 WD40 domain. VP-containing peptides were previously identified based upon a core signature motif E-S-D-E-x-x-x-V-P-[E/D]-Φ-G, where Φ designated a hydrophobic residue (Holm *et al*, 2001; Uljon *et al*, 2016). Our structural analyses of a diverse set of VP-containing peptides now reveal that COP1 has evolved a highly plastic VP-binding pocket, which enables sequence-divergent VP motifs from different plant light signaling components to compete with each other for COP1 binding. It is reasonable to assume that many more *bona fide* VP

**Figure 5.  COP1$^{Lys422Ala}$ shows a delayed flowering phenotype under long days and suggests that CO–CRY2 may compete for the COP1 VP-binding site.**

A   Representative image of wild type (Ws), *cop1-5/Pro₃₅ₛ:YFP-COP1*, *cop1-5/Pro₃₅ₛ:YFP-COP1$^{Lys422Ala}$*, *cop1-5/Pro₃₅ₛ:YFP-COP1$^{Tyr441Ala}$*, and *cop1-5/Pro₃₅ₛ:YFP-COP1$^{Trp467Ala}$* transgenic lines grown for 39 days in long-day conditions.

B   Representative image of individual wild type (Ws), *co-11*, *cop1-5/Pro₃₅ₛ:YFP-COP1*, and *cop1-5/Pro₃₅ₛ:YFP-COP1$^{Lys422Ala}$* plants grown for 39 days in long-day conditions.

C   Quantification of flowering time. Means and SD are shown ($n = 14$).

D   Number of rosette and cauline leaves at flowering. Means and SD are shown ($n = 14$).

E   Immunoblot analysis of CRY2 and actin (loading control) protein levels in wild type (Col), *cry2-1*, wild type (Ws), *cop1-5/Pro₃₅ₛ:YFP-COP1*, and *cop1-5/Pro₃₅ₛ:YFP-COP1$^{Lys422Ala}$* seedlings grown for 4 days in darkness.

F   ITC assays between the CO VP peptide (left) and CRY2 VP peptide (right) versus the COP1 WD40 and the COP1$^{Lys422Ala}$ WD40 domains, with a table summarizing their corresponding affinities. The following concentrations were typically used (titrant into cell): CO–COP1 (2,240 µM in 195 µM); CO–COP1$^{Lys422Ala}$ (1,200 µM in 120 µM); CRY2–COP1 (750 µM in 70 µM); and CRY2–COP1$^{Lys422Ala}$ (3,000 µM in 175 µM). The stoichiometries of binding (N) for CO–COP1 = 1.03 $\pm$ 0.00; CO–COP1$^{Lys422Ala}$ = 0.89 $\pm$ 0.03; CRY2–COP1 = 1.04 $\pm$ 0.20; CRY2–COP1$^{Lys422Ala}$ = 1 (fixed) (all measurements $\pm$ standard deviation).

G, H   Binding kinetics of the (G) full-length activated CRY2 and (H) CRY2$^{NC80}$ versus the COP1 WD40 domain obtained by GCI experiments. Sensorgrams of protein injected are shown in red, with their respective heterogenous ligand binding model fits in black. The following amounts were typically used: ligand—COP1 (2,000 pg/mm²); analyte—CRY2 (highest concentration 14 µM), CRY2$^{NC80}$ (highest concentration 450 µM). $k_a$ = association rate constant, $k_d$ = dissociation rate constant, $K_d$ = dissociation constant.

Source data are available online for this figure.

motifs may exist and our structures now provide sequence fingerprints to enable their bioinformatic discovery.

Interestingly, although we predict that at least some of our COP1 mutant variants (e.g., Trp467Ala) completely disrupt the interaction

with VP motif harboring COP1 targets, all COP1 variants can complement the *cop1-5* seedling-lethal phenotype and largely the *cop* phenotype in darkness (Holm *et al*, 2001; and this work, Fig EV1D and E). This could imply that a significant part of COP1

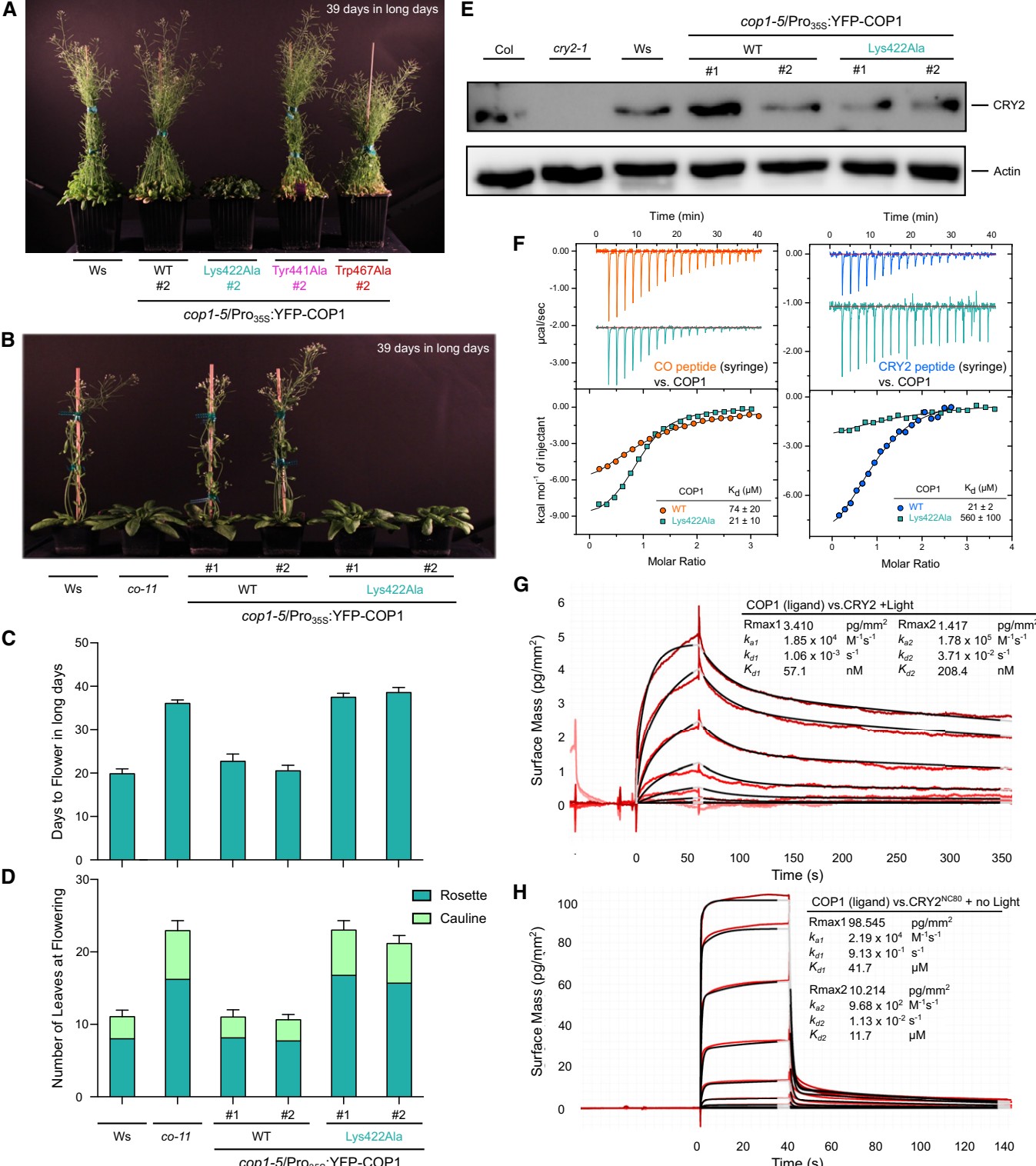

**Figure 5.**

activity is independent from the VP-mediated destabilization of photomorphogenesis-promoting transcription factors. It has been recently suggested that part of the *cop1* phenotype could be explained by COP1-mediated stabilization of PIFs (Pham *et al*, 2018). Our COP1 lines could be used to gain further insight into this aspect of COP1 activity.

Human COP1 prefers to bind phosphorylated substrates, and their post-translational regulation may also be relevant in plants (Hardtke *et al*, 2000; Uljon *et al*, 2016). In this respect, it is noteworthy that the full-length COP1 protein may exist as an oligomer as well as in complex with other light signaling proteins, such as SPA proteins (Seo *et al*, 2003; Huang *et al*, 2013; Sheerin *et al*, 2015; Holtkotte *et al*, 2017). The four SPA protein family members share a similar domain architecture with COP1, consisting of an N-terminal kinase-like domain, a central coiled-coil domain, and a C-terminal WD40 domain ($\sim 45\%$ amino acid identity with the COP1 WD40 domain), and are partially redundant in their activities (Yang & Wang, 2006; Ordoñez-Herrera *et al*, 2015). Mutations in the SPA1 WD40 domain residues Lys767 and Trp812, which correspond to COP1 residues Lys422 and Trp467, cannot complement the *spa1-3* mutant (Yang & Wang, 2006). These higher-order complexes are known to be part of some but not all light signaling pathways and could thus encode additional determinants for signaling specificity (Hoecker, 2017; Podolec & Ulm, 2018). In addition to the competition mechanism presented here, it has been observed that active cryptochrome and phytochrome receptors directly interact with SPA proteins and thereby separate COP1 from SPA proteins, which results in COP1 inactivation (Lian *et al*, 2011; Liu *et al*, 2011; Zuo *et al*, 2011; Lu *et al*, 2015; Sheerin *et al*, 2015). However, early UVR8 signaling is independent of SPA proteins (Oravecz *et al*, 2006) and may thus rely exclusively on the competition mechanism described here. For cryptochrome signaling, the VP-mediated competition and COP1-SPA disruption mechanisms are obviously not mutually exclusive but likely function in parallel *in vivo* to reinforce COP1-SPA E3 ligase inactivation in blue-light signaling. Reconstitution of a photoreceptor–COP1/SPA signaling complex may offer new insights into these different targeting mechanisms in the future.

# Materials and Methods

## Cell culture, strains, plasmids, and antibodies

### Sf9 cell culture

*Spodoptera frugiperda* Sf9 cells (Thermo Fisher) were cultured in Sf-4 Baculo Express insect cell medium (Bioconcept).

### Yeast strains

The following *Saccharomyces cerevisiae* reporter strains were used: L40 (*MAT**a** trp1 leu2 his3 ade2 LYS2::lexA-HIS3 URA3::lexA-lacZ GAL4*) (Vojtek & Hollenberg, 1995), Y190 (*MAT**a** ura3-52 his3-200 lys2-801 ade2-101 trp1-901 leu2-3 112 gal4Δ gal80Δ cyh^r2 LYS2:: GAL1$_{UAS}$-HIS3$_{TATA}$-HIS3 MEL1 URA3::GAL1$_{UAS}$-GAL1$_{TATA}$-lacZ*), and Y187 (*MATα ura3-52 his3-200 ade2-101 trp1-901 leu2-3 112 gal4Δ met⁻ gal80Δ URA3::GAL1$_{UAS}$-GAL1$_{TATA}$-lacZ MEL1*) (Yeast Protocols Handbook, Clontech).

### Plants

*cop1-4* (Oravecz *et al*, 2006) *cop1-5* (McNellis *et al*, 1994), *cop1-5/ Pro$_{35S}$:YFP-COP1*, *cop1-5/Pro$_{35S}$:YFP-COP1$^{Lys422Ala}$*, *cop1-5/Pro$_{35S}$: YFP-COP1$^{Tyr441Ala}$*, *cop1-5/Pro$_{35S}$:YFP-COP1$^{Trp467Ala}$* (this work), *uvr8-7* (Favory *et al*, 2009), *uvr8-7/Pro$_{35S}$:UVR8$^{HY5C44}$*, *uvr8-7/ Pro$_{35S}$:UVR8$^{HY5VP}$*, and *uvr8-7/Pro$_{35S}$:UVR8$^{TRIB1}$* (this work) are in the *Arabidopsis thaliana* Wassilewskija (Ws) accession. The *cry2-1* (Guo *et al*, 1998) mutant is in the Columbia accession. The *co-11* allele was generated in the Ws accession (this work) using CRISPR/ Cas9 technology (Wang *et al*, 2015).

### Plasmids

All constructs and plasmids used are listed as described below and in the Appendix Table S1.

### Antibodies

All antibodies used are listed as described below and in the Appendix Table S1.

## Protein expression and purification

All COP1, UVR8, HY5, and CRY2 proteins were produced as follows: The desired coding sequence was PCR amplified (see Appendix Table S1 for primers) or NcoI/NotI digested from codon-optimized genes (GeneArt or Twist Biosciences) for expression in Sf9 cells. Chimeric UVR8 constructs were PCR amplified directly from vectors used for yeast 3-hybrid assays (see below). All constructs except CRY2$^{NC80}$ were cloned into a modified pFastBac (Geneva Biotech) insect cell expression vector, via NcoI/NotI restriction enzyme sites or by Gibson assembly (Gibson *et al*, 2009). The modified pFastBac vector contains a tandem N-terminal His$_{10}$-Twin-Strep-tags followed by a TEV (tobacco etch virus protease) cleavage site. CRY2$^{NC80}$ was cloned into a modified pET-28 a (+) vector (Novagen) containing a tandem N-terminal His$_{10}$-Twin-Strep-tags followed by a TEV (tobacco etch virus protease) cleavage site by Gibson assembly. Mutagenesis was performed using an enhanced plasmid mutagenesis protocol (Liu & Naismith, 2008).

pFastBac constructs were transformed into DH10MultiBac cells (Geneva Biotech), white colonies indicating successful recombination were selected, and bacmids were purified by the alkaline lysis method. Sf9 cells were transfected with the desired bacmid with Profectin (AB Vector). eYFP-positive cells were observed after 1 week and subjected to one round of viral amplification. Amplified P2 virus (between 5 and 10% culture volume) was used to infect Sf9 cells at a density between $1–2 \times 10^6$ cells/ml. Cells were incubated for 72 h at 28°C before the cell pellet was harvested by centrifugation at $2,000 \times g$ for 20 min and stored at −20°C.

CRY2$^{NC80}$ was produced in transformed Rosetta (DE3) pLysS (Novagen) cells. *Escherichia coli* were grown in 2xYT broth with kanamycin. One liter of broth was inoculated with 20 ml of a saturated overnight pre-culture, grown at 37°C until OD600 $\sim 0.5$, induced with IPTG at a final concentration of 0.2 mM, and then shaken for another 16 h at 18°C. The cell pellet was harvested by centrifugation at $2,000 \times g$ for 20 min and stored at −20°C.

Every 1 l of Sf9 or bacterial cell culture was dissolved in 25 ml of Buffer A (300 mM NaCl, 20 mM HEPES 7.4, 2 mM BME), supplemented with glycerol (10% v/v), 5 µl turbonuclease, and 1

Roche cOmplete Protease Inhibitor tablets. Dissolved pellets were lysed by sonication, and insoluble materials were separated by centrifugation at $60,000 \times g$ for 1 h at 4°C. The supernatant was filtered through tandem 1- and 0.45-μm filters before $Ni^{2+}$-affinity purification (HisTrap excel, GE Healthcare). $Ni^{2+}$-bound proteins were washed with Buffer A and eluted directly into a coupled Strep-Tactin Superflow XT column (IBA) by Buffer B (500 mM NaCl, 500 mM imidazole pH 7.4, 20 mM HEPES pH 7.4). Twin-Strep-tagged-bound proteins on the Strep-Tactin column were washed with Buffer A and eluted with 1× Buffer BXT (IBA). Proteins were cleaved overnight at 4°C with TEV protease. Cleaved proteins were subsequently purified from the protease and affinity tag by a second $Ni^{2+}$-affinity column or by gel filtration on a Superdex 200 Increase 10/300 GL column (GE Healthcare). Proteins were concentrated to 3–10 mg/ml and either used immediately or aliquoted and frozen directly at −80°C. Typical purifications were from 2 to 5 l of cell pellet.

All protein concentrations were measured by absorption at 280 nm and calculated from their molar extinction coefficients. Molecular weights of all proteins were confirmed by MALDI-TOF mass spectrometry. SDS–PAGE gels to assess protein purity are shown in Appendix Fig S7.

For UVR8 monomerization and activation by UV-B, proteins were diluted to their final assay concentrations (as indicated in the figure legends) in Eppendorf tubes and exposed to 60 min at max intensity (69 mA) under UV-B LEDs (Roithner Lasertechnik GmbH) on ice.

For CRY2 activation, proteins were diluted to their final assay concentrations (as indicated in the figure legends) in Eppendorf tubes and exposed to 10 min of fluorescent light on ice.

### Analytical size-exclusion chromatography

Gel filtration experiments were performed using a Superdex 200 Increase 10/300 GL column (GE Healthcare) pre-equilibrated in 150 mM NaCl, 20 mM HEPES 7.4, and 2 mM BME. 500 μl of the respective protein solution or a mixture (~ 4 μM per protein) was loaded sequentially onto the column, and elution at 0.75 ml/min was monitored by UV absorbance at 280 nm.

### Isothermal titration calorimetry (ITC)

All experiments were performed in a buffer containing 150 mM NaCl, 20 mM HEPES 7.4, and 2 mM BME. Peptides were synthesized and delivered as lyophilized powder (Peptide Specialty Labs GmbH) and dissolved directly in buffer. The peptides were centrifuged at $14,000 \times g$ for 10 min, and only the supernatant was used. The dissolved peptide concentrations were calculated based upon their absorbance at 280 nm and their corresponding molar extinction coefficient. Typical experiments consisted of titrations of 20 injections of 2 μl of titrant (peptides) into the cell containing COP1 at a 10-fold lower concentration. Typical concentrations for the titrant were between 500 and 3,000 μM for experiments depending on the affinity. Experiments were performed at 25°C and a stirring speed of 1,000 rpm on an ITC200 instrument (GE Healthcare). All data were processed using Origin 7.0 and fit to a one-site binding model after background buffer subtraction.

### Grating-coupled interferometry (GCI)

The Creoptix WAVE system (Creoptix AG), a label-free surface biosensor, was used to perform GCI experiments. All experiments were performed on 2PCH or 4PCH WAVEchips (quasi-planar poly-carboxylate surface; Creoptix AG). After a borate buffer conditioning (100 mM sodium borate pH 9.0, 1 M NaCl; Xantec), COP1 (ligand) was immobilized on the chip surface using standard amine-coupling: 7 min activation [1:1 mix of 400 mM *N*-(3-dimethylamino-propyl)-*N′*-ethylcarbodiimide hydrochloride and 100 mM *N*-hydroxysuccinimide (both Xantec)], injection of COP1 (10 μg/ml) in 10 mM sodium acetate pH 5.0 (Sigma) until the desired density was reached, and final quenching with 1 M ethanolamine pH 8.0 for 7 min (Xantec). Since the analyte CRY2 showed non-specific binding on the surface, BSA (0.5% in 10 mM sodium acetate, pH 5.0; BSA from Roche) was used to passivate the surface between the injection of COP1 and ethanolamine quenching. For a typical experiment, the analyte (UVR8/CRY2) was injected in a 1:3 dilution series (highest concentrations as indicated in the figure legends) in 150 mM NaCl, 20 mM HEPES 7.4, and 2 mM BME at 25°C. Blank injections were used for double referencing and a dimethyl sulfoxide (DMSO) calibration curve for bulk correction. Analysis and correction of the obtained data were performed using the Creoptix WAVEcontrol software (applied corrections: X and Y offset; DMSO calibration; double referencing), and a one-to-one binding model or a heterogenous ligand model with bulk correction was used to fit all experiments.

### Protein crystallization and data collection

Crystals of co-complexes of $HY5^{39–48}$–$COP1^{349–675}$, $UVR8^{406–413}$–$COP1^{349–675}$, $HY5^{39–48}$–$COP1^{349–675,\ Lys422Ala}$, $UVR8^{406–413}$–$COP1^{349–675,\ Lys422Ala}$, and $COL3^{287–294}$–$COP1^{349–675}$ were grown in sitting drops and appeared after several days at 20°C when 5 mg/ml of COP1 supplemented with three- to 10-fold molar excess in peptide was mixed with twofold (v/v) more mother liquor (1:2 ratio; protein:buffer) containing 2 M $(NH_4)_2SO_4$ and 0.1 M HEPES pH 7.4 or 0.1 M Tris pH 8.5. Crystals were harvested and cryoprotected in mother liquor supplemented with 25% glycerol and frozen under liquid nitrogen.

Crystals of complexes of $HYH^{27–34}$–$COP1^{349–675}$, $HFR1^{57–64}$–$COP1^{349–675}$, $STO^{240–247}$–$COP1^{349–675}$, and $CRY1^{544–552}$–$COP1^{349–675}$ were grown in sitting drops and appeared after several days at 20°C when 5 mg/ml of COP1 supplemented with three- to 10fold molar excess in peptide was mixed with twofold (v/v) more mother liquor (1:2 ratio; protein:buffer) containing 1.25 M sodium malonate pH 7.5. Crystals were harvested and cryoprotected in mother liquor supplemented with 25% glycerol and frozen under liquid nitrogen.

All datasets were collected at beam line PX-III of the Swiss Light Source, Villigen, Switzerland. Native datasets were collected with λ = 1.03 Å. All datasets were processed with XDS (Kabsch, 1993) and scaled with AIMLESS as implemented in the CCP4 suite (Winn *et al*, 2011).

### Crystallographic structure solution and refinement

The structures of all the peptide–COP1 WD40 complexes were solved by molecular replacement as implemented in the program

Phaser (McCoy *et al*, 2007), using PDB-ID 5IGO as the initial search model. The final structures were determined after iterative rounds of model-building in COOT (Emsley & Cowtan, 2004), followed by refinement in REFMAC5 (Murshudov *et al*, 2011) as implemented in CCP4 and phenix.refine (Adams *et al*, 2010). Polder omit maps were generated for the UVR8[406–413]–COP1 structure by omitting residue Tyr407 of the bound peptide as implemented in phenix.polder. Final statistics were generated as implemented in phenix.table_one. All figures were rendered in UCSF Chimera (Pettersen *et al*, 2004).

## Plant transformation

To generate the *cop1-5*/Pro[35S]:YFP-COP1 line, COP1 cloned into pENTR207C was introduced into the Gateway-compatible binary vector pB7WGY2 (Karimi *et al*, 2002). COP1 mutated versions were generated by PCR-based site-directed mutagenesis, cloned into pDONR207, and then introduced in pB7WGY2 (Karimi *et al*, 2002). The wild type version of the construct contains an additional Gateway-cloning-related 14 amino acids' linker sequence between the YFP and COP1. *cop1-5* heterozygous plants (kan[R]) were transformed using the floral dip method (Clough & Bent, 1998). Lines homozygous for the *cop1-5* mutation and for single locus insertions of the Pro[35S]:YFP-COP1 transgene were selected.

To generate lines expressing chimeric UVR8 receptors, the *HY5* and *TRIB1* sequences were introduced by PCR to the *UVR8* coding sequences as indicated, and the chimeras were cloned into the Gateway-compatible binary vector pB2GW7 (Karimi *et al*, 2002) for transformation into the *uvr8-7* mutant background. Lines homozygous with single genetic locus transgene insertions were selected.

To generate a *co* mutant in the Ws background, designated *co-11*, plants were transformed with the CRISPR/Cas9 binary vector pHEE401E (Wang *et al*, 2015) in which an sgRNA specific to the *CO* CDS was inserted (see Appendix Table S1). A plant was isolated in T2 and propagated, harboring a 1 base-pair insertion after the codon for residue Asp137 leading to a frameshift and a premature stop codon after four altered amino acids (*DPRGR**; *D* representing Asp137 in CO, * representing the premature stop).

## Plant growth conditions

For experiments at seedling stage, *Arabidopsis* seeds were surface-sterilized and sown on half-strength MS medium (Duchefa), stratified in the dark at 4°C for 48 h, and grown under aseptic conditions in controlled light conditions at 21°C. For hypocotyl length and anthocyanin measurements, the MS medium was supplemented with 1% sucrose (AppliChem). For flowering experiments, *Arabidopsis* plants were grown on soil in long-day (16 h/8 h; light/dark cycles) growth chambers at 21°C.

UV-B treatments were performed as described before, using Osram L18W/30 tubes, supplemented with narrowband UV-B from Philips TL20W/01RS tubes (Oravecz *et al*, 2006; Favory *et al*, 2009).

## Hypocotyl length assays

For hypocotyl length measurements, at least 60 seedlings were randomly chosen, aligned, and scanned. Measurements were performed using the NeuronJ plugin of ImageJ (Meijering *et al*,

2004). Violin and box plots were generated using the ggplot2 library in R (Wickham, 2009).

## Anthocyanin quantification

Accumulation of anthocyanin pigments was assayed as described previously (Yin *et al*, 2012). In brief, 40–60 mg of seedlings were harvested, frozen, and grinded before adding 250 µl acidic methanol (1% HCl). Samples were incubated on a rotary shaker for 1 h, the supernatant was collected, and absorbances at 530 and 655 nm were recorded using a spectrophotometer. Anthocyanin concentration was calculated as $(A_{530} - 2.5 * A_{655})/mg$, where mg is the fresh weight of the sample.

## Protein extraction and immunoblotting

For total protein extraction, plant material was grinded and incubated with an extraction buffer composed of 50 mM Na-phosphate pH 7.4, 150 mM NaCl, 10% (v/v) glycerol, 5 mM EDTA, 0.1% (v/v) Triton X-100, 1 mM DTT, 2 mM $Na_3VO_4$, 2 mM NaF, 1% (v/v) Protease Inhibitor Cocktail (Sigma), and 50 µM MG132, as previously described (Arongaus *et al*, 2018).

Proteins were separated by electrophoresis in 8% (w/v) SDS–polyacrylamide gels and transferred to PVDF membranes (Roth) according to the manufacturer's instructions (iBlot dry blotting system, Thermo Fisher Scientific), except for CRY2 immunoblots, which were transferred on nitrocellulose membranes (Bio-Rad).

For protein gel blot analyses, anti-UVR8[426–440] (Favory *et al*, 2009), anti-UVR8[1–15] (Yin *et al*, 2015), anti-UVR8[410–424] (Heijde & Ulm, 2013), anti-GFP (Living Colors® A.v. Monoclonal Antibody, JL-8; Clontech), anti-actin (A0480; Sigma-Aldrich), and anti-CRY2[588–602] (Eurogentec, raised against the peptide N′-CEGKNLE-GIQDSSDQI-C′ and affinity purified) were used as primary antibodies. Horseradish peroxidase-conjugated anti-rabbit and anti-mouse immunoglobulins (Dako) were used as secondary antibodies. Signal detection was performed using the ECL Select Western Blotting Detection Reagent (GE Healthcare) and an Amersham Imager 680 camera system (GE Healthcare).

## Quantitative real-time PCR

RNA was extracted from seedlings using the RNeasy Plant Mini Kit (Qiagen) following the manufacturer's instructions. RNA samples were treated for 20 min with RNA-free DNAse (Qiagen) followed by addition of DEPC-treated EDTA for inactivation at 65°C for 10 min. Reverse transcription was performed using TaqMan Reverse Transcription Reagents (Applied Biosystems), using a 1:1 mixture of oligo-dT and random hexamer primers. Quantitative real-time PCR was performed on a QuantStudio 5 Real-Time PCR system (Thermo Fisher Scientific) using PowerUp SYBR Green Master Mix reagents (Applied Biosystems). Gene-specific primers for *CHS*, *COP1*, *ELIP2*, *HY5*, *RUP2*, and *UVR8* were described before (Favory *et al*, 2009; Gruber *et al*, 2010; Heijde *et al*, 2013); *18S* expression was used as reference gene (Vandenbussche *et al*, 2014); and expression values were calculated using the $\Delta\Delta C_t$ method (Livak & Schmittgen, 2001) and normalized to the wild type. Each reaction was performed in technical triplicates; data shown are from three biological repetitions.

## Flowering time assays

For quantitative flowering time measurements, the number of days to flowering was determined at bolting, and rosette and cauline leaf numbers were counted when the inflorescence reached approximately 1 cm in length (Möller-Steinbach *et al*, 2010).

## Yeast 2-hybrid and 3-hybrid assays

For yeast 2-hybrid assay, *COP1* and its mutated variants were introduced into pGADT7-GW (Marrocco *et al*, 2006; Yin *et al*, 2015) and *HY5*, *UVR8*, and *UVR8^C44* were introduced into pBTM116-D9-GW (Stelzl *et al*, 2005; Yin *et al*, 2015; Binkert *et al*, 2016). Vectors were co-transformed into the L40 strain (Vojtek & Hollenberg, 1995) using the lithium acetate-based transformation protocol (Gietz, 2014). Transformants were selected and grown on SD/-Trp/-Leu medium (Formedium). For the analysis of β-galactosidase activity, enzymatic assays using chlorophenol red-β-D-galactopyranoside (Roche Applied Science) as substrate were performed as described (Yeast Protocols Handbook, Clontech).

For yeast 3-hybrid analysis, pGADT7-GW-COP1 was transformed into the Y190 strain (Harper *et al*, 1993). *HY5*, *HYH,* and *HFR1^N186* were cloned into the BamHI/EcoRI site of pBridge (Clontech), and *UVR8*, *UVR8^ValPro/AlaAla*, *UVR8^1–396*, and *UVR8^HY5C44* were cloned into the BglII/PstI cloning site, followed by transformation into the Y187 strain (Harper *et al*, 1993). Transformants were mated, selected, and grown on SD/-Trp/-Leu/-Met medium (Formedium). For the analysis of β-galactosidase activity, filter lift assays were performed as described (Yeast Protocols Handbook, Clontech). Enzymatic assays using chlorophenol red-β-D-galactopyranoside (Roche Applied Science) were performed as described (Yeast Protocols Handbook, Clontech). For repression of $Pro_{Met25}$:UVR8 expression, SD/-Trp/-Leu/-Met medium was supplemented with 1 mM L-methionine (Fisher Scientific).

For assays, yeast cells were grown for 2 days at 30°C in darkness or under narrow-band UV-B (Philips TL20W/01RS; 1.5 μmol/m$^2$/s), as indicated.

## Quantification and statistical analysis

Data of ITC and GCI binding assays are reported with errors as indicated in their figure legends.

# Data availability

The atomic coordinates of complexes have been deposited with the following Protein Data Bank (PDB) accession codes: HY5$^{39–48}$–COP1$^{349–675}$: **6QTO** (https://www.rcsb.org/structure/6qto), UVR8$^{406–413}$–COP1$^{349–675}$: **6QTQ** (https://www.rcsb.org/structure/6qtq), HY5–COP1$^{349–675,Lys422Ala}$: **6QTR** (https://www.rcsb.org/structure/6qtr), UVR8–COP1$^{349–675,Lys422Ala}$: **6QTS** (https://www.rcsb.org/structure/6qts), HYH$^{27–34}$–COP1$^{349–675}$: **6QTT** (https://www.rcsb.org/structure/6qtt), STO$^{240–247}$–COP1$^{349–675}$: **6QTU** (https://www.rcsb.org/structure/6qtu), HFR1$^{57–64}$–COP1$^{349–675}$: **6QTV** (https://www.rcsb.org/structure/6qtv), CRY1$^{544–552}$–COP1$^{349–675}$: **6QTW** (https://www.rcsb.org/structure/6qtw), and COL3$^{287–294}$–COP1$^{349–675}$: **6QTX** (https://www.rcsb.org/structure/6qtx).

**Expanded View** for this article is available online.

## Acknowledgements

This work was supported by an HHMI International Research Scholar Award to M.H., the Swiss National Science Foundation (grant number 31003A_175774 to R.U.), and the European Research Council (ERC) under the European Union's Seventh Framework Programme (grant no. 310539 to R.U.). R.P. was supported by an iGE3 PhD Salary Award, and K.L. was supported by an EMBO Long-term Fellowship (ALTF 493-2015). We thank Luis Lopez-Molina for technical assistance with the UV-B LEDs, Fabio Spiga for GCI analyses, and Darwin for growing a crystal seeding tool.

## Author contributions

KL, RP, RU, and MH designed experiments and wrote the article. KL performed all protein expression and purification, ITC, GCI and X-ray crystallography experiments and their analyses. RP with the help of RC generated all plant lines, and performed all *in vivo* plant and yeast experiments and their analyses.

## Conflict of interest

The authors declare that they have no conflict of interest.

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
