## [Review Process File · The EMBO Journal]

Plant photoreceptors and their signaling components compete for COP1 binding via VP-peptide motifs

Kelvin Lau, Roman Podolec, Richard Chappuis, Roman Ulm, and Michael Hothorn

Review timeline:

Submission date:	29th Mar 2019
Editorial Decision:	6th May 2019
Revision received:	29th May 2019
Accepted:	7th Jun 2019

Editor: Ieva Gailite

Transaction Report:

1st Editorial Decision

6th May 2019

Thank you for submitting your manuscript for consideration by the EMBO Journal. We have now received three referee reports on your manuscript, which are included below for your information.

As you will see from the comments, all reviewers appreciate the work and the quality of the data and recommend publication of the manuscript after a minor revision. Given these positive evaluations from three experts of the field, I would like to invite you to submit a revised version of your manuscript, in which you address the minor comments from all reviewers.

REFeree REPORTS:

Referee #1:

The ms by Lau et al describes an excellent study to explain the competitive interactions among photoreceptors, especially UVR8, COP1, and transcription factors. The authors first solved UVR8-COP1 and HY5-COP1 (domains) complexes, which support the hypothesis that the VP motif of HY5 and UVR8 interact with COP1. They then systematically analyzed the affinities of those interactions, the physiological effects of the interacting residues, showing the cooperative binding between the two domains of UVR8 and that only photoactivated UVR8 could effectively compete with HY5 for COP1. The authors then demonstrated that the VP motif of other COP1 substrates rescue the activity of VP-defective UVR8 mutant *in vivo*, using the chimeric mutant UVR8 fused to VP motifs of HY5 or a human COP1 substrate TRIB1. They then showed that most known of the 7 or more COP1-interacting proteins interact with the WD domain of COP1 via their sequence divergent VP motif with comparable affinity in ITC assay and the similar configuration in the respective crystal structures analyzed. More interestingly, these authors showed that cop1(K422A) has a reduced affinity to CRY2, whereas NC80 of CRY2 has the similar affinity to COP1 as the VP motif of CRY2 (part of NC80). These observations appear to nicely explain why cop1(K422A) mutant causes late-flowering phenotype and loss of CO protein as well as the previously puzzling observation that a mere 80 residue of CRY2 is more active than the 612-residue full length CRY2.

In summary, this reviewer finds this ms representing an excellent study of the photosignaling mechanism, with important questions, well-designed experiments, redundant assays for key results, enormous workload, exciting findings, and well-written ms. Although this reviewer wishes the authors would clarify whether CRY2-COP1 and CRY2-cop1(K422A) interactions are blue light-dependent on their hand (previously reported to be blue light-independent by others), this ms can basically be published in its present form.

Referee #2:

GENERAL SUMMARY

COP1 (CONSTITUTIVELY PHOTOMORPHOGENIC 1) is a key regulator of light responses in plants. Plants lacking functional COP1 induce light responses in the dark, including inhibition of hypocotyl growth, apical hook unfolding, cotyledon opening, anthocyanin biosynthesis, and expression of light responses marker genes. COP1 is part of an E3 ubiquitin ligase complex and responsible for binding target proteins. Transcription factors that promote light responses, such as HY5, HFR1, CO, and others are targets of the COP1 containing E3 ubiquitin ligase complex and they bind to COP1. COP1 dependent degradation of these transcription factors represses light responses. In plants exposed to light, different photoreceptors bind to COP1 and inhibit the E3 ubiquitin ligase activity, which allows COP1 targets to accumulate and induce light responses. Photoreceptors binding to COP1 include the UV-B receptor UVR8, the blue light receptors cryptochrome 1 and 2 (CRY1, CRY2) and the red/far-red light sensing phytochromes. However, how exactly these photoreceptors inhibit COP1 activity leading to stabilisation of HY5, HFR1, CO, and other COP1 targets is unknown. Many COP1 targets contain short sequence-divergent Val-Pro (VP) motifs that are required for binding to COP1; also UVR8 and cryptochromes contain such motifs. In their work, Lau and co-authors investigated binding of COP1 to COP1 target proteins and UVR8 using crystal structures of a COP1 fragment bound to the VP motifs of COP1 target proteins and UVR8. They found that COP1 target proteins and UVR8 bind to the same site in COP1. Furthermore, they used quantitative binding assays to measure binding affinities of COP1 target proteins and UVR8 for COP1. The binding affinities observed for HY5, inactive UVR8 and UVR8 activated by UV-B light suggest that only active but not inactive UVR8 can efficiently compete with HY5 for binding to COP1, suggesting that light activated UVR8 stabilises HY5 by preventing it from binding to COP1/displacing it from COP1. These findings are supported by yeast three hybrid competition assays and reverse genetic experiments with Arabidopsis plants expressing different COP1 mutant versions. Finally, the authors show data suggesting that - similar to UVR8 - also CRY1 and CRY2 use a VP motif-dependent mechanism to compete with transcription factors for COP1 binding.

This is an excellent manuscript addressing a fundamental question in photobiology. The strength of this manuscript is that the authors use methods that have not been used in the past to investigate how photoreceptors inactivate COP1 and thereby allow COP1 targets to accumulate. The findings are a major step forward towards understanding at the molecular/structural level how photoreceptors regulate gene expression.

The conclusions are well supported by experimental evidence and the manuscript is well written. I have only a couple of minor comments and suggestions.

MAJOR CONCERNS

none

MINOR COMMENTS

Line 111, 127; Fig. S4B, C: COP1 W467A does not bind to UVR8 and HY5 derived peptides but still complements the dark phenotype of cop1-5. If COP1 W467A is impaired in binding to

substrates, I would expect that it does not complement the dark phenotype of *cop1-5*. Do the authors think that COP1 W467A still binds to other substrates that would promote photomorphogenesis and that are still targeted for degradation by COP1 W467A? I am also not sure if COP1 W467A fully complements the dark phenotype of *cop1-5*; in Fig. S4B the hooks appear unfolded and the cotyledons opened (especially line 2).

Line 127; Fig. 1F: The authors write that all lines expressed comparable levels of the YFP-fusion. I know that it can be hard to obtain lines expressing comparable levels and I am sure the authors did their best to obtain such lines. Thus, I accept that expression levels are different but I don't think they should be described as "comparable".

Line 134; Fig. 1E, G, H: I agree that the lines expressing COP1 Y441A have elongated hypocotyls compared to the WT and that this suggests enhanced COP1 activity. However, COP1 Y441A does not bind UVR8 but the lines expressing COP1 Y441A still respond to UV-B light with strongly reduced hypocotyl growth. How do the authors explain this?

Line 140: The authors write that lines expressing COP1 Y441A are impaired in UV-B signalling. However, Fig. 1G, H shows that this line strongly responds to UV-B light and that hypocotyl length of UV-B treated seedlings is reduced to about 50% of the hypocotyl length of the non-treated seedlings. This 50% reduction is similar to the WT; the main difference between WT and COP1 Y441A expressing seedlings is that seedlings expressing COP1 Y441A have generally longer hypocotyls than the WT or seedlings expressing WT COP1. Can the authors comment on that?

Line 137; Fig. 1I, J; Fig. S4A: Seedlings expressing COP1 Y441A clearly respond to UV-B light regarding hypocotyl growth but they do not or only weakly respond to UV-B treatment regarding expression of marker genes. Can the authors suggest an explanation for this?

Line 204: "... can bind COP1 affinities comparable to wild-type (Figs 3G and S10) and are dimers in vitro that monomerize under UV-B ..." → please check if this sentence is correct.

Line 170, 172: Fig. instead of Figs.

Line 198; Fig. 1H: The *uvr8-7* mutant is a null mutant but it appears to respond to UV-B light in Fig. 1H (not as much as the WT but hypocotyl growth is reduced in *uvr8-7* treated with UV-B). Can the authors comment on this?

Figure legend Fig. 1E: I think stoichiometry is not shown.

Figure legend Fig. 5F: I think stoichiometry is not shown.

NON-ESSENTIAL SUGGESTIONS

none

Referee #3:

This paper provides intriguing new information regarding the interaction of plant and human COP1 with diverse partner proteins, in particular the Arabidopsis UVR8 UV/B photoreceptor and HY5 transcription factor, but also including CO, HYH, HRF1, COL3, STO and human TRIB1. In a technical tour de force, the authors use transgenic plants to complement null mutants with wild-type and mutated/substituted components, yeast 2-hybrid (and 3-hybrid, which is non-trivial), size-exclusion chromatography, grating-coupled interferometry, and isothermal titration calorimetry to analyse protein-protein interactions. In the case of COP1-UVR8 they could show the particular action of UV-absorption by the propeller tryptophans on the interactions and the likely role of UV-dependent competition between HY5 and UVR8 for binding. On top of all this, they used X-ray diffraction to solve the structures of numerous COP1-oligopeptide co-crystals at very high resolution (1.1 - 1.4 and 1.9 Å), their refined models yielding excellent R_{work} and R_{free} statistics. Although one could hardly ask for more, nit-picking remains possible. In particular, I would have

liked to see the appropriate negative (dark) control for the COP1-CRY2 interaction to accompany Fig. 5 G & H. Also, several of the GCI models deviate significantly from the measured kinetics although this is not discussed.

The study underlines the astonishing structural and functional similarities of the COP1 from the two kingdoms of life and provides interesting insights into the interactions of COP1 to the rather dissimilar "VP" binding motifs of the ligands. The data also show that the propeller interactions act cooperatively in these associations. By combining *in vitro* and *in planta* methods, the authors contribute substantially to our knowledge of the roles of COP1 in general and its interactions with the VP motif in particular. Although still incomplete, the story is quite remarkable.

Some trivial aspects:

L95: "can directly sense" - "sense" is too close to sentient understanding, "directly interacts with" would be better.

L109: "explain" rather than "rationalize".

L154: "independent" is an adjective, it should be the adverb.

L225 begins with a split infinitive

L263: COP1 is not a photoreceptor and therefore cannot sense light.

Figures: The k_a and k_d association and dissociation rate constants are given as e.g. " $5.24e-2$ ", that is using so-called scientific notation. The authors might take enough care to give the values conventionally (e.g. $5.24 \cdot 10^{-2}$)

Jon Hughes

1st Revision - authors' response

29th May 2019

We have prepared a revised version of our manuscript EMBOJ-2019-102140R "Plant photoreceptors compete with COP1 substrates for COP1 binding.", which addresses the reviewer comments and your editorial suggestions. Please find a detailed point-by-point response below. In addition changes in the manuscript are highlighted in yellow.

Referee #1:

The ms by Lau et al describes an excellent study to explain the competitive interactions among photoreceptors, especially UVR8, COP1, and transcription factors. The authors first solved UVR8-COP1 and HY5-COP1 (domains) complexes, which support the hypothesis that the VP motif of HY5 and UVR8 interact with COP1. They then systematically analyzed the affinities of those interactions, the physiological effects of the interacting residues, showing the cooperative binding between the two domains of UVR8 and that only photoactivated UVR8 could effectively compete with HY5 for COP1. The authors then demonstrated that the VP motif of other COP1 substrates rescue the activity of VP-defective UVR8 mutant *in vivo*, using the chimeric mutant UVR8 fused to VP motifs of HY5 or a human COP1 substrate TRIB1. They then showed that most known of the 7 or more COP1-interacting proteins interact with the WD domain of COP1 via their sequence divergent VP motif with comparable affinity in ITC assay and the similar configuration in the respective crystal structures analyzed. More interestingly, these authors showed that cop1(K422A) has a reduced affinity to CRY2, whereas NC80 of CRY2 has the similar affinity to COP1 as the VP motif of CRY2 (part of NC80). These observations appear to nicely explain why cop1(K422A) mutant causes late-flowering phenotype and loss of CO protein as well as the previously puzzling observation that a mere 80 residue of CRY2 is more active than the 612-residue full length CRY2. In summary, this reviewer finds this ms representing an excellent study of the photosignaling mechanism, with important questions, well-designed experiments, redundant assays for key results, enormous workload, exciting findings, and well-written ms.

OUR RESPONSE

We would like to thank reviewer #1 for their positive comments.

Although this reviewer wishes the authors would clarify whether CRY2-COP1 and CRY2-cop1(K422A) interactions are blue light-dependent on their hand (previously reported to be blue light-independent by others), this ms can basically be published in its present form.

OUR RESPONSE

We would like to emphasize the fact that in our hands, only full-length CRY2 with additional fluorescent light supplemented was able to interact with COP1 (Fig 5G and newly added Appendix Fig S8, see updated materials and methods). The respective statement in the manuscript reads (lines 257-259): “ In line with this, we find that only recombinant light-activated full-length CRY2 binds wild-type COP1 with nanomolar affinity in quantitative GCI experiments (Fig 5G and Appendix Fig S8).”

Referee #2:

GENERAL SUMMARY

COP1 (CONSTITUTIVELY PHOTOMORPHOGENIC 1) is a key regulator of light responses in plants. Plants lacking functional COP1 induce light responses in the dark, including inhibition of hypocotyl growth, apical hook unfolding, cotyledon opening, anthocyanin biosynthesis, and expression of light responses marker genes. COP1 is part of an E3 ubiquitin ligase complex and responsible for binding target proteins. Transcription factors that promote light responses, such as HY5, HFR1, CO, and others are targets of the COP1 containing E3 ubiquitin ligase complex and they bind to COP1. COP1 dependent degradation of these transcription factors represses light responses. In plants exposed to light, different photoreceptors bind to COP1 and inhibit the E3 ubiquitin ligase activity, which allows COP1 targets to accumulate and induce light responses. Photoreceptors binding to COP1 include the UV-B receptor UVR8, the blue light receptors cryptochrome 1 and 2 (CRY1, CRY2) and the red/far-red light sensing phytochromes. However, how exactly these photoreceptors inhibit COP1 activity leading to stabilization of HY5, HFR1, CO, and other COP1 targets is unknown. Many COP1 targets contain short sequence-divergent Val-Pro (VP) motifs that are required for binding to COP1; also UVR8 and cryptochromes contain such motifs. In their work, Lau and co-authors investigated binding of COP1 to COP1 target proteins and UVR8 using crystal structures of a COP1 fragment bound to the VP motifs of COP1 target proteins and UVR8. They found that COP1 target proteins and UVR8 bind to the same site in COP1. Furthermore, they used quantitative binding assays to measure binding affinities of COP1 target proteins and UVR8 for COP1. The binding affinities observed for HY5, inactive UVR8 and UVR8 activated by UV-B light suggest that only active but not inactive UVR8 can efficiently compete with HY5 for binding to COP1, suggesting that light activated UVR8 stabilises HY5 by preventing it from binding to COP1/displacing it from COP1. These findings are supported by yeast three hybrid competition assays and reverse genetic experiments with Arabidopsis plants expressing different COP1 mutant versions. Finally, the authors show data suggesting that - similar to UVR8 - also CRY1 and CRY2 use a VP motif-dependent mechanism to compete with transcription factors for COP1 binding. This is an excellent manuscript addressing a fundamental question in photobiology. The strength of this manuscript is that the authors use methods that have not been used in the past to investigate how photoreceptors inactivate COP1 and thereby allow COP1 targets to accumulate. The findings are a major step forward towards understanding at the molecular/structural level how photoreceptors regulate gene expression.

OUR RESPONSE

We thank the reviewer for these enthusiastic comments.

The conclusions are well supported by experimental evidence and the manuscript is well written. I have only a couple of minor comments and suggestions.

Line 111, 127; Fig. S4B, C: COP1 W467A does not bind to UVR8 and HY5 derived peptides but still complements the dark phenotype of cop1-5. If COP1 W467A is impaired in binding to substrates, I would expect that it does not complement the dark phenotype of cop1-5. Do the authors think that COP1 W467A still binds to other substrates that would promote photomorphogenesis and that are still targeted for degradation by COP1 W467A? I am also not not sure if COP1 W467A fully complements the dark phenotype of cop1-5; in Fig. S4B the hooks appear unfolded and the cotyledons opened (especially line 2).

OUR RESPONSE

Thank you for pointing this out to us. Indeed the respective lines show only partial complementation, displaying shorter hypocotyls and open cotyledons (previous Fig. S4B,C, now EV1D, E). As you are well aware, *cop1-5* null mutants are seedling lethal, but *cop1-4*, in which the VP-interacting WD40 domain is truncated, is viable (due to the activity of SPA proteins, Ordonez-Herrera et al., Mol. Plant, 2015)). Thus plants with impaired COP1-VP binding may still be viable. We have revised our manuscript to highlight that our COP1 W467A mutant may not be impaired in interaction with all substrates and still target other photomorphogenesis-promoting factors for degradation, or retains an activity that is independent of the VP-interaction motif. The revised statement reads (lines 301-303): “Likewise, partial complementation of COP1^{Trp467Ala} reveals that it may not be impaired in interacting with all COP1 substrates and that it may still target other photomorphogenesis-promoting factors for degradation or retains an activity that is independent of the VP-peptide binding pocket”

Line 127; Fig. 1F: The authors write that all lines expressed comparable levels of the YFP-fusion. I know that it can be hard to obtain lines expressing comparable levels and I am sure the authors did their best to obtain such lines. Thus, I accept that expression levels are different but I don't think they should be described as "comparable".

OUR RESPONSE

Line 128-129: we have deleted “comparable levels of”.

Line 134; Fig. 1E, G, H: I agree that the lines expressing COP1 Y441A have elongated hypocotyls compared to the WT and that this suggests enhanced COP1 activity. However, COP1 Y441A does not bind UVR8 but the lines expressing COP1 Y441A still respond to UV-B light with strongly reduced hypocotyl growth. How do the authors explain this?

Line 140: The authors write that lines expressing COP1 Y441A are impaired in UV-B signalling. However, Fig. 1G, H shows that this line strongly responds to UV-B light and that hypocotyl length of UV-B treated seedlings is reduced to about 50% of the hypocotyl length of the non-treated seedlings. This 50% reduction is similar to the WT; the main difference between WT and COP1 Y441A expressing seedlings is that seedlings expressing COP1 Y441A have generally longer hypocotyls than the WT or seedlings expressing WT COP1. Can the authors comment on that? Line 137; Fig. 1I, J; Fig. S4A: Seedlings expressing COP1 Y441A clearly respond to UV-B light regarding hypocotyl growth but they do not or only weakly respond to UV-B treatment regarding expression of marker genes. Can the authors suggest an explanation for this?

OUR RESPONSE

We agree with reviewer #2 that COP1 Y441A mutant plants show inhibition of hypocotyl elongation, a longer-term phenotype after 4 days UV exposure. This phenotype is likely due to the recently described direct interaction of UVR8 with transcription factors that are required for hypocotyl elongation (WRKY36, BES1, BIM1) (Liang et al., Dev Cell 2018; Yang et al., Nat Plants 2018). UVR8 negatively regulates these transcription factors, a UVR8 activity that is independent of COP (Liang et al., Dev Cell 2018; Yang et al., Nat Plants 2018). Moreover, we cannot exclude any role of the UVR8 interaction with COP1 that is not dependent on the UVR8 C-terminal VP domain (see Yin et al., Plant Cell 2015, former Figs. 2D, S6B,C, S7B, now Fig. 2D, Appendix Fig S4B,C and Fig. EV2). This may affect COP1^{Tyr441Ala} activity sufficiently enough to observe a long-term phenotype as hypocotyl growth inhibition. Importantly, however, transcriptional responses after 2h of UV-B treatment are clearly abolished (former Figs. 1I, J and S4A, B, now Figs. 1H, I and EV1B, C). This response is an early read-out of UVB signaling, which has been previously linked to UVR8-COP1-HY5 signaling (e.g. Favory et al, 2009; Binkert et al, 2014).

We discuss this COP1-independent UVR8 signaling pathway contributing to hypocotyl growth inhibition in more detail in our revised manuscript. We have added a statement to our results section (lines 136-141): “Importantly, in contrast to YFP-COP1, transcriptional responses for UV-B-induced marker genes like *HY5*, *RUP2*, *ELIP2* and *CHS* are clearly abolished in YFP-COP1^{Lys422Ala}, YFP-COP1^{Tyr441Ala} or YFP-COP1^{Trp467Ala} after 2h of UV-B treatment (Figs 1H and I, and EV1B and C). These responses represent an early read-out of UV-B signaling, which has been previously linked to UVR8-COP1-HY5 (Brown et al., 2005, Oravecz et al., 2006, Favory et al., 2009; Binkert et al., 2014).”

And the discussion section now reads (lines 293-300): “It is interesting to note that the COP1^{Tyr441Ala} mutant still shows UV-B-dependent hypocotyl growth inhibition even though the COP1^{Tyr441Ala} protein is impaired in its interaction with UVR8. This UV-B response is possibly due to a recently

described COP1-independent UVR8 activity involving direct interaction and inhibition of transcription factors that promote hypocotyl elongation (Liang et al, 2018; Yang et al, 2018b). Moreover, we cannot exclude that the UVR8 interaction with COP1 that is independent of the UVR8 C-terminal VP domain (Fig 2D, Appendix Fig S4B and Fig EV2B; see also Yin *et al*, 2015) can affect COP1^{Tyr441Ala} activity sufficiently to see a long-term phenotype such as hypocotyl growth inhibition.”

Line 204: "... can bind COP1 affinities comparable to wild-type (Figs 3G and S10) and are dimers *in vitro* that monomerize under UV-B ..." → please check if this sentence is correct.

OUR RESPONSE

We have clarified this statement that now reads (lines 207-210): “In line with this, the UVR8^{HY5C44} chimera can displace HY5 from COP1 in yeast 3-hybrid assays (Fig 3F). All the UVR8 chimeras can bind COP1 with affinities comparable to wild-type (Figs 3G and EV3) and are dimers *in vitro* that monomerize under UV-B light (Fig 3H and Appendix Fig S7D).”

Line 170, 172: Fig. instead of Figs.

OUR RESPONSE

Corrected.

Line 198; Fig. 1H: The *uvr8-7* mutant is a null mutant but it appears to respond to UV-B light in Fig. 1H (not as much as the WT but hypocotyl growth is reduced in *uvr8-7* treated with UV-B). Can the authors comment on this?

OUR RESPONSE

In this experiment, seedlings were grown under different light fields, with or without supplemental UV-B tubes. A minor effect on hypocotyl elongation can sometimes be detected and it is not clear what the exact reason is. Next to a slight possible difference in the visible light between the two treatments, it may also indicate hypocotyl shortening due to UV-B stress (note that *uvr8* mutants do not acclimate during the 4 days of UV-B exposure and may thus start to show hypocotyl growth inhibition due to UV-B stress, as has been documented before), or the existence of a presently unknown UV-B response mechanism. Nonetheless, it is important to note that UV-B signaling is absent in these *uvr8* mutant seedlings: UV-B induced anthocyanin accumulation (Fig. 3D) and induction of UV-B-responsive gene expression, in addition to *HY5* in Fig. 1H, are clearly absent in *uvr8* mutants (please see Figs. 1H, I, 3E, EV1 B,C). Also please refer to Fig. 3 for additional replicates.

Figure legend Fig. 1E: I think stoichiometry is not shown. Figure legend Fig. 5F: I think stoichiometry is not shown.

OUR RESPONSE

The binding stoichiometries have been included in the revised figure legends.

Referee #3:

This paper provides intriguing new information regarding the interaction of plant and human COP1 with diverse partner proteins, in particular the Arabidopsis UVR8 UV/B photoreceptor and HY5 transcription factor, but also including CO, HYH, HRF1, COL3, STO and human TRIB1. In a technical tour de force, the authors use transgenic plants to complement null mutants with wild-type and mutated/substituted components, yeast 2-hybrid (and 3-hybrid, which is non-trivial), size-exclusion chromatography, grating-coupled interferometry, and isothermal titration calorimetry to analyse protein-protein interactions. In the case of COP1-UVR8 they could show the particular action of UV-absorption by the propeller tryptophans on the interactions and the likely role of UV-dependent competition between HY5 and UVR8 for binding. On top of all this, they used X-ray diffraction to solve the structures of numerous COP1-oligopeptide co-crystals at very high resolution (1.1 - 1.4 and 1.9 Å), their refined models yielding excellent R_{work} and R_{free} statistics.

OUR RESPONSE

We thank this reviewer for appreciating our work.

Although one could hardly ask for more, nit-picking remains possible. In particular, I would have liked to see the appropriate negative (dark) control for the COP1-CRY2 interaction to accompany Fig. 5 G & H.

OUR RESPONSE:

We have added a control experiment, in which no additional light was supplemented to the CRY2 protein before the GCI binding assay (newly added Appendix Figure S8).

Also, several of the GCI models deviate significantly from the measured kinetics although this is not discussed.

OUR RESPONSE:

Thank you for pointing out this important issue. We have re-analyzed our data with a specialist from Creoptix, and together we came up with two possible scenarios that could explain the observed deviation of the raw sensograms from the 1:1 kinetic model fit. (1) As outlined in our manuscript, COP1 may cooperatively bind UV-B light-activated UVR8 using two distinct, but possibly overlapping binding sites (one for the UVR8 core and one for its VP-peptide motif). (2) Upon COP1 (ligand) capture to the immobilized surface there could be two distinct populations that affect how it interacts with its analyte binding partners. To account for these two possibilities, we have now reanalyzed the respective GCI experiment using a heterogenous ligand model, where COP1 represents the ligand. Compared to the 1:1 fits, the new heterogenous ligand model fits result in one population of ligands (>75-90% of the total signal; Rmax1) having similar kinetics and dissociation constants as previously obtained from 1:1 kinetic model fits. A small fraction of the immobilized ligands (Rmax2) also contributes to the overall signal with dissociation constants that are also comparable to the ones from Rmax1 and to the previous 1:1 model fits. The new analysis is shown in revised Fig. 5, with the derived kinetic parameters being in full agreement with initial hypothesis.

The study underlines the astonishing structural and functional similarities of the COP1 from the two kingdoms of life and provides interesting insights into the interactions of COP1 to the rather dissimilar "VP" binding motifs of the ligands. The data also show that the propeller interactions act cooperatively in these associations. By combining in vitro and in planta methods, the authors contribute substantially to our knowledge of the roles of COP1 in general and its interactions with the VP motif in particular. Although still incomplete, the story is quite remarkable. Some trivial aspects: L95: "can directly sense" - "sense" is too close to sentient understanding, "directly interacts with" would be better.

OUR RESPONSE

Changed as requested.

L109: "explain" rather than "rationalize".

OUR RESPONSE

Changed as requested.

L154: "independent" is an adjective, it should be the adverb.

OUR RESPONSE

Corrected.

L225 begins with a split infinitive

OUR RESPONSE

Corrected.

L263: COP1 is not a photoreceptor and therefore cannot sense light.

OUR RESPONSE:

We have modified this sentence, it now reads (lines 269-270): "The COP1 E3 ubiquitin ligase is a central signaling hub that integrates inputs from plant light sensing photoreceptors."

Figures: The k_a and k_d association and dissociation rate constants are given as e.g. "5.24e-2", that is using so-called scientific notation. The authors might take enough care to give the values conventionally (e.g. $5.24 \cdot 10^{-2}$)

Jon Hughes

OUR RESPONSE

Corrected throughout.

2nd Editorial Decision

7th Jun 2019

Thank you very much for incorporating the requested changes in the revised manuscript. I am now pleased to inform you that your manuscript has been accepted for publication in The EMBO Journal. Congratulations on a nice study!

Corresponding Author Name: Michael Hothorn

Manuscript Number: EMBOJ-2019-102140R